# FieryGS: In-the-Wild Fire Synthesis with Physics-Integrated Gaussian Splatting

**Qianfan Shen[1]\*, Ningxiao Tao[2]\*, Qiyu Dai[2]\*†, Tianle Chen[3], Minghan Qin[4], Yongjie Zhang[4], Mengyu Chu[2]‡, Wenzheng Chen[5,6]‡, Baoquan Chen[2]‡**

[1]School of EECS, Peking University
[2]School of Intelligence Science and Technology, Peking University
[3]Yuanpei College, Peking University
[4]ByteDance Seed
[5]Wangxuan Institute of Computer Technology, Peking University
[6]Beijing Academy of Artificial Intelligence, Beijing, China

## Abstract

We consider the problem of synthesizing photorealistic, physically plausible combustion effects in in-the-wild 3D scenes. Traditional CFD and graphics pipelines can produce realistic fire effects but rely on handcrafted geometry, expert-tuned parameters, and labor-intensive workflows, limiting their scalability to the real world. Recent scene modeling advances like 3D Gaussian Splatting (3DGS) enable high-fidelity real-world scene reconstruction, yet lack physical grounding for combustion. To bridge this gap, we propose FieryGS, a physically-based framework that integrates physically-accurate and user-controllable combustion simulation and rendering within the 3DGS pipeline, enabling realistic fire synthesis for real scenes. Our approach tightly couples three key modules: (1) multimodal large-language-model-based physical material reasoning, (2) efficient volumetric combustion simulation, and (3) a unified renderer for fire and 3DGS. By unifying reconstruction, physical reasoning, simulation, and rendering, FieryGS removes manual tuning and automatically generates realistic, controllable fire dynamics consistent with scene geometry and materials. Our framework supports complex combustion phenomena—including flame propagation, smoke dispersion, and surface carbonization—with precise user control over fire intensity, airflow, ignition location and other combustion parameters. Evaluated on diverse indoor and outdoor scenes, FieryGS outperforms all comparative baselines in visual realism, physical fidelity, and controllability. Project page can be found at `https://pku-vcl-geometry.github.io/FieryGS/`.

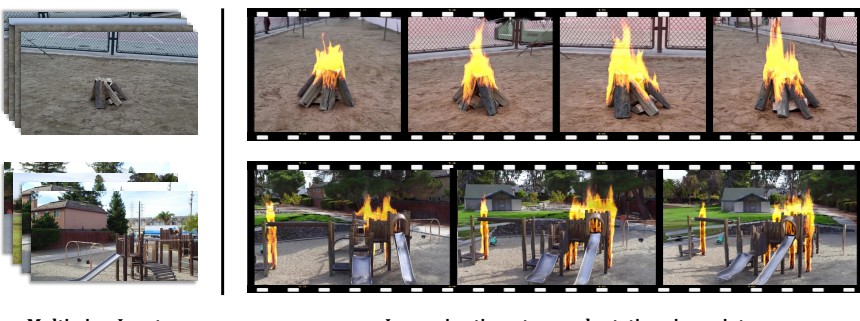

**Multi-view Inputs** ← **Increasing time steps and rotating viewpoints** →

Figure 1: FieryGS synthesizes physically-grounded fire effects from multi-view image, enabling controllable and realistic fire for in-the-wild scenes.

---

\*Joint first authors
†Project lead
‡Corresponding authors

# 1 INTRODUCTION

Synthesizing realistic and controllable combustion effects grounded in in-the-wild 3D scenes is critical for applications ranging from AR/VR, gaming, and film production to virtual fire drills, heritage preservation, and robotics perception under adverse conditions, where fire must be visually convincing, physically plausible, interactively controllable, and well-aligned with the real world. Existing approaches, however, fall short of meeting these requirements (Table 1).

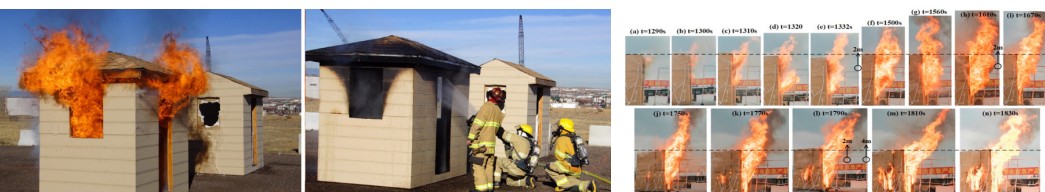

Figure 2: Left: Real-world combustion in a live-fire drill (5280 Fire Science); Right:Full-scale combustion test measuring flame spread time (Zhang et al., 2021)

Table 1: Applicability comparison of combustion approaches. FieryGS offers accessible fire simulation for real-world scenes by combining scene-aligned physics, visual fidelity, efficiency, and user control.

| Method | Real-world Applicability | Visual Fidelity | Physical Fidelity | Parameter Control | User Friendliness | Scalability |
|---|---|---|---|---|---|---|
| Full-scale Experiments | ✓ | ✓ | ✓ | ✗ | ✗ | ✗ |
| CFD Methods | ✗ | sim-to-real gap | ✓ | ✓ | expert-only | ✗ |
| VFX Tools | ✗ | sim-to-real gap | ✓ | ✓ | expert-only | ✗ |
| Commercial Software | ✓ | ✗ | pre-stored | ✗ | ✓ | ✗ |
| Large Video Models | ✓ | ✓ | ✗ | ✗ | ✓ | ✓ |
| FieryGS (Ours) | ✓ | ✓ | ✓ | ✓ | ✓ | ✓ |

Notes: **Real-world Applicability** indicates ease of use in real scenes, where CFD/VFX requires manual modeling. **Visual Fidelity** measures perceptual realism, where CFD/VFX suffer sim-to-real gaps and commercial software overlays pre-computed results. **Physical Fidelity** checks consistency with physics, where large video models are data-driven and commercial software uses pre-stored effects. **Parameter Control** reflects the ability to vary conditions, where full-scale experiments are costly to repeat, large video models offer little precise control, and commercial software is limited to pre-stored effects. **User Friendliness** considers usability, where full-scale experiments are dangerous and CFD/VFX requires experts. **Scalability** is automatic adaptation to new scenes at low cost. Full-scale experiments are expensive, CFD/VFX needs manual modeling, and commercial software is limited to pre-stored effects.

The most authentic option—full-scale fire experiments, such as burning life-sized structures (Fig. 2)—is prohibitively expensive, risky, and irreproducible, making systematic exploration under varying conditions infeasible. Alternatively, digital approaches like computational fluid dynamics (CFD) or visual effect (VFX) software (e.g., Houdini, Blender) incorporate physics-based simulation but depend on asset construction, detailed material annotation, carefully discretized geometry, and brittle simulation–rendering pipelines (Lakkonen, 2024; Mahadika & Utami, 2025). Thus, targeting real-world scenes demands impractical manual specification, and each step remains an incomplete approximation, inevitably producing a pronounced sim-to-real gap that limits practical deployment. With the rise of Large Video Models (LVM), it has become possible to add fire effects directly to footage, but the results lack physical consistency and precise controllability. Due to these limitations, current commercial software (SimsUshare, 2025; Digital Combustion, 2025) instead relies on overlaying pre-stored fire effects onto scenes, without ensuring physical fidelity.

Recent advances in scene modeling present new opportunities. Methods such as Neural Radiance Fields (NeRF)(Mildenhall et al., 2020) and 3DGS(Kerbl et al., 2023) enable high-fidelity 3D reconstruction from multi-view images, providing highly detailed surface information with strong real-world alignment. Although primarily designed for static appearance capture, their visual fidelity and rendering efficiency suggest potential for further material inference and physics-informed modeling. Some prior works leverage such reconstructions to incorporate physical properties (Li et al.; Cai et al., 2024; Li et al., 2023; Feng et al., 2024; Dai et al., 2025; Hsu et al., 2024) to model related phenomena such as fluid dynamics or deformable objects. However, realistic combustion remains out of reach, as it requires accurate scene-level material inference, complex simulation tightly coupled with scene representation, and fine-grained controllability over fire behavior.

To bridge this gap, we introduce FieryGS, a physically based framework that integrates accurate and controllable combustion simulation into the 3DGS pipeline. Our method automatically generates

photorealistic, dynamic fire in reconstructed scenes while allowing precise control over fire intensity, airflow, ignition location, and other parameters. The framework tightly couples three components:

- Multimodal-large-language-model(MLLM)-based material reasoning, zero-shot inferring combustion-relevant reliable properties from 3DGS reconstructions;
- Controllable volumetric combustion simulation with wood charring via a principled balance of computational cost and visual realism;
- A novel unified renderer, combining fire, smoke, and 3DGS for seamless photorealistic emission and illumination.

Tightly coupling these modules enables realistic fire effects to emerge directly from real-world data without expert design or handcrafted inputs. FieryGS is, to our knowledge, the first framework that generates visually and physically realistic combustion in in-the-wild scenes, while being efficient and supporting precise user controls over ignition location, fire intensity, airflow, and other parameters. Experiments across tabletop, indoor, and outdoor scenarios show that FieryGS outperforms state-of-the-art baselines in visual realism, physical fidelity, and user controllability, advancing fire synthesis from labor-intensive, expert-heavy workflows to automatic, real-world aligned process.

## 2 RELATED WORK

**Challenges in Combustion Simulation**   Combustion simulation has long been studied in both CFD and computer graphics, with physically based models developed to replicate fire behavior (Husain & Srivastava, 2018; Nguyen et al., 2002; Nielsen et al., 2022; Feldman et al., 2003; Kwatra et al., 2010), material changes such as pyrolysis and charring of wood (Liu et al., 2024a), and volumetric rendering of flames and smoke (Huang et al., 2014; Nguyen et al., 2002; Pegoraro & Parker, 2006). While these methods excel in specific aspects, they rely heavily on manual inputs, such as detailed geometry and material properties, and often require expert knowledge to combine multiple tools, resulting in limited flexibility and sim-to-real gaps in diversity and fidelity. Existing commercial software (SimsUshare, 2025; Digital Combustion, 2025) supports real-world case studies but relies on pre-stored fire effects, lacking both physical consistency and control over fire parameters. These limitations motivate a combustion framework that can automatically align with real-world scenes while maintaining efficiency, controllability, and physical accuracy.

**Neural Scene Representations for Physically-Grounded Editing**   Recent NeRF and 3DGS representations have enabled high-fidelity 3D reconstruction and inspired extensions to physical property inference. Some estimate parameters like Young's modulus, fluid viscosity, friction or stiffness from videos (Li et al.; Cai et al., 2024; Cao et al., 2024; Zhong et al., 2024), while others (Zhang et al., 2024; Huang et al., 2024a; Liu et al., 2024b; Lin et al., 2025; Liu et al., 2025) exploit dynamics in video models to infer material properties. LLMs provide a complementary direction to physical property reasoning, as in NeRF2Physics (Zhai et al., 2024), GaussianProperty (Xu et al., 2024), and PUGS (Shuai et al., 2025). However, they remain object-centric and do not address combustion-related attributes. Parallel efforts integrate explicit simulation with neural representations, including deformable bodies via Material Point Method (MPM) (Xie et al., 2024; Zhang et al., 2024; Huang et al., 2024a; Liu et al., 2024b), weather phenomena (Li et al., 2023), fluid–solid interactions (Feng et al., 2024), and rainfall (Dai et al., 2025). AutoVFX (Hsu et al., 2024) supports flame effects using Blender's built-in physics, but its dynamics are driven by LLM-generated scripts rather than spatiotemporal physical interactions, lacking physical consistency and control. We address this gap by introducing the first framework that integrates combustion simulation with 3DGS, enabling controllable and physically faithful fire synthesis.

## 3 METHODS

Given multi-view images, we reconstruct 3DGS scenes and infer combustion properties through zero-shot MLLM reasoning (Sec. 3.1). The properties guide a physics-based combustion simulation (Sec. 3.2), which is rendered together with the scene using unified volumetric rendering (Sec. 3.3). Fig. 3 illustrates the pipeline.

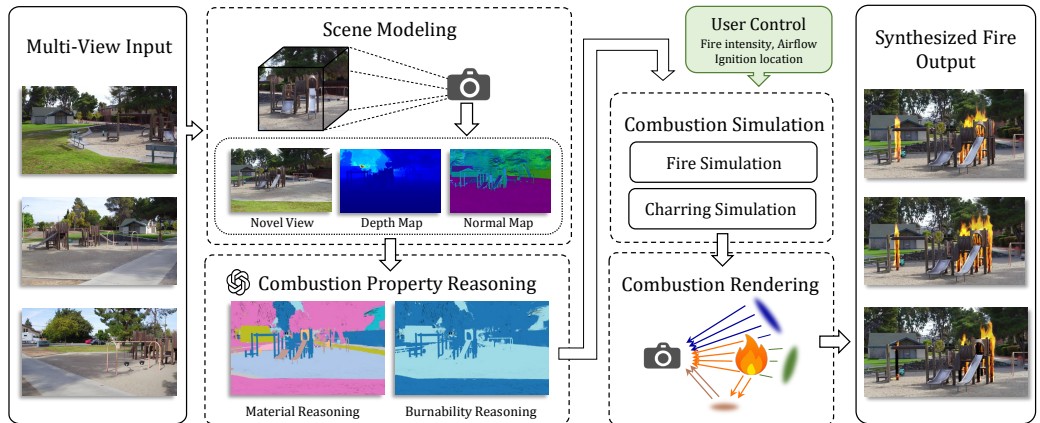

Figure 3: **Overall Pipeline of FieryGS.** Given multi-view images as input, we first apply PGSR (Chen et al., 2024) to reconstruct scenes with high-quality normal and depth. Next, we leverage MLLM to infer combustion-related properties, such as material type and burnability. Based on these, we conduct combustion simulations, enabling fire and charring effects with user control. A unified volumetric renderer seamlessly integrates 3DGS and fire, accounting for smoke scattering, fire illumination, and charring, producing realistic fire results.

## 3.1 Scene Modeling with Combustion Property Reasoning

High-fidelity 3D modeling of appearance, geometry, and physical properties in in-the-wild scenes is essential for realistic combustion simulation. We adopt PGSR (Chen et al., 2024), a recent 3DGS-based method that jointly reconstructs photorealistic appearance and accurate geometry, for scene reconstruction. To enable physically plausible fire simulation, we estimate combustion-relevant material properties for each Gaussian in reconstructed 3DGS, including material type, burnability, thermal diffusivity, and smoke color. Recent MLLMs have shown strong capabilities in inferring material from 2D images. However, extending the capabilities to in-the-wild 3DGS scenes remains challenging. Intuitively, nearby Gaussians with visual similarities are likely to share same material properties. Inspired by recent work in 3DGS segmentation (Ye et al., 2024; Cen et al., 2025), we first partition Gaussians into coherent 3D regions, each with a shared material. Then, each region is rendered to 2D and passed to an MLLM for material inference. To ensure reliable MLLM prediction, inference is performed from the viewpoint where the target 3D region has the highest visibility.

**Preliminary of 3D Gaussian Splatting**   3DGS (Kerbl et al., 2023) models a scene as a set of anisotropic Gaussians, each parameterized by its center, covariance ($\Sigma$), opacity, and view-dependent color encoded with spherical harmonics. During rendering, a Gaussian is projected into screen space with covariance $\Sigma' = JV\Sigma V^\top J^\top$, where $V$ is the camera extrinsic matrix and $J$ the Jacobian of the projection. Pixel colors are obtained by alpha blending over depth-sorted Gaussians.

**3D Gaussian Segmentation**   Given a reconstructed 3DGS model, we first assign each Gaussian a learnable feature vector $f_g \in \mathbb{R}^D$, where $D$ is the feature dimension. These features are rendered into 2D feature maps via 3DGS alpha blending. We then apply SAM (Kirillov et al., 2023), a foundation model for 2D segmentation, to obtain segmentation maps across multiple views. Following SAGA (Cen et al., 2025), we adopt contrastive learning to train the feature vectors $f_g$, encouraging pixels within the same mask to share similar embeddings. After training, Gaussians associated with the same 3D region exhibit similar features. We then apply HDBSCAN algorithm (McInnes et al., 2017) to cluster these feature vectors into instance-level 3D segments, each assumed to correspond to a distinct material region (See Appendix A.1.1 for hyperparameter details).

**MLLM-based Combustion Property Reasoning**   For each segmented region in 3D Gaussians, we rasterize it into 2D and perform material inference using an MLLM. In real-world scenes, complex occlusions cause large visibility differences across viewpoints, and limited exposure to the target region can degrade MLLM prediction accuracy. To address this, we select the viewpoint where the target 3D region has the highest visibility, determined by counting the number of unoccluded Gaussians based on rendered depth maps. We then feed GPT-4o (Hurst et al., 2024) a set of 2D renderings from the selected viewpoint, along with a tailored prompt (see Appendix A.1.2), to infer the material type and combustion-relevant physical properties. The predicted attributes are projected back to 3D by directly assigning to all Gaussians in the corresponding region. On average, GPT-4o

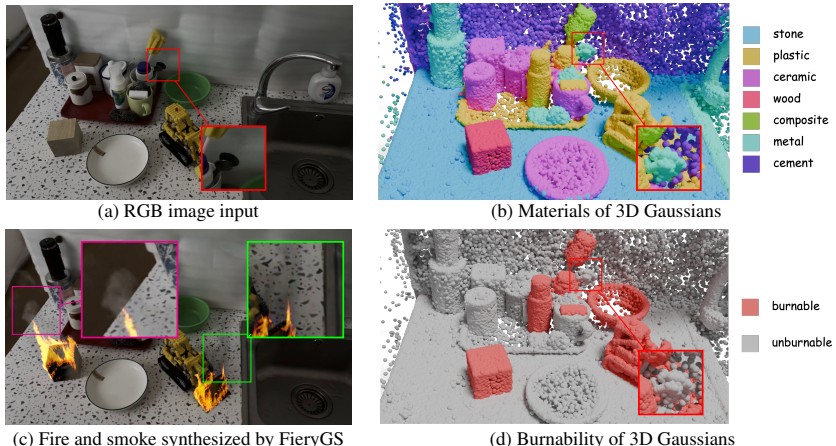

(a) RGB image input        (b) Materials of 3D Gaussians

(c) Fire and smoke synthesized by FieryGS      (d) Burnability of 3D Gaussians

Figure 4: **Combustion Property Reasoning.** Given an RGB input (a), our method reliably predicts material types (b) and burnability (d). In a complex region with metal spoons inside a mug surrounded by various materials, the method distinguishes the spoons and correctly infers their non-flammable metallic nature. These results drive the combustion simulation and rendering, where material-specific behaviors are applied—for instance, combustion produces white smoke for the wooden box and black smoke for the plastic Lego (c).

API calls cost about \$0.55 per scene, making our pipeline highly economical (see Appendix B.2). We further validate the robustness and accuracy of the material reasoning results (see Appendix B.3).

The result is a 3DGS augmented with physical and combustion-aware attributes (Fig. 4). An occupancy grid is then constructed, where a voxel is labeled as occupied if it overlaps with one or more 3D Gaussians whose opacity exceeds a given threshold, and further labeled as combustible if any of these Gaussians are burnable. This grid defines the domain for combustion simulation, with unoccupied voxels representing air regions and occupied voxels representing solid regions.

## 3.2 Combustion Simulation

Given the occupancy grid obtained in Section 3.1, we run combustion simulation in two parts. Fire simulation computes combustion state in air regions, which are then used for flame and smoke rendering. Charring simulation updates combustible regions with the degree of charring, supporting the rendering of charred surfaces. Focusing on efficiency, our method simplifies processes that have little visual impact, striking a balance between computational performance and visual fidelity. Compared to CFD and VFX methods, which require manual geometry modeling and explicit specification of combustible regions, our pipeline leverages scene modeling and material reasoning to automatically initialize geometry, infer material properties, and identify combustible areas. Meanwhile, users retain flexible control over key parameters, making the simulation workflow largely automated and easy to customize. In the following, we present fire simulation, charring simulation, and user control, while further implementation details are provided in Appendix A.2.

**Fire Simulation** We model flame dynamics using the following equations:

$$\frac{\partial \mathbf{u}}{\partial t} + \mathbf{u} \cdot \nabla \mathbf{u} = -\frac{1}{\rho} \nabla p + \mathbf{f}, \quad \text{s.t. } \nabla \cdot \mathbf{u} = 0; \quad \frac{\partial Y}{\partial t} + \mathbf{u} \cdot \nabla Y = -k. \tag{1}$$

where $\mathbf{u}$ is the divergence-free velocity field, $\rho$ and $p$ denote density and pressure, and $Y$ is the reaction coordinate variable ($Y = 1$ for burning material, $Y = 0$ for unburnt material). At the beginning of the simulation, all voxels are initialized with $Y = 0$. Only the voxels corresponding to user-specified ignition points, which are also predicted as combustible in the occupancy grid, are set to $Y = 1$, indicating the onset of combustion.

In this formulation, we choose an incompressible flow model (Nguyen et al., 2002) to balance physical plausibility with computational simplicity, in contrast to compressible formulations (Liu et al., 2024a) that provide higher physical fidelity but at the cost of greater complexity. Among the external forces $\mathbf{f}$ in Eq. 1, we consider buoyancy force $\mathbf{f}_{buo} = \alpha(T - T_{air})\mathbf{z}$ and vorticity confinement force $\mathbf{f}_{vor}$ (Nguyen et al., 2002). To further improve efficiency, the temperature $T$ is approximated as a quadratic function of reaction coordinate variable $Y$, rather than solved through PDE-based thermal

models (Nguyen et al., 2002; Nielsen et al., 2022). This simplification makes the simulation pipeline more concise while still capturing the correlation between combustion progress and temperature.

**Charring Simulation**  For combustible solids, we simulate temperature evolution by solving a simplified heat transfer equation:

$$\frac{\partial T_m}{\partial t} = \beta \nabla^2 T_m + \gamma_m(T_{amb}^4 - T_m^4) + S_{T_m}, \tag{2}$$

where $T_m$ denotes the material temperature, $\beta$ is the thermal diffusivity, and $\gamma_m$ is the radiative cooling coefficient. To avoid the high cost of explicitly modeling internal heat generation, $S_{T_m}$ is approximated by clamping $T_m$ to $T_{burn}$ once the ignition threshold $T_{ign}$ is exceeded. Based on the simulated temperature, the relative char mass is computed as $\frac{\partial M_c}{\partial t} = \varepsilon_c \xi(T_m)$, where $M_c$ denotes the relative char mass, with $M_c = 1$ representing a fully charred state and $M_c = 0$ indicating the opposite. The parameter $\varepsilon_c$ represents the charring rate, while $\xi(T_m)$ equals 1 if $T_m \geq T_{ign}$ and 0 otherwise. Unlike prior work that incorporates more detailed mechanisms such as insulation-layer formation or volatile release (Liu et al., 2024a), our formulation deliberately omits these processes. This simplification makes the simulation more efficient while still capturing the visually dominant aspects of charring. Subsequently, each 3D Gaussians directly inherits the $M_c$ value from its containing grid voxel, providing a simple mapping to guide charring visualization.

**User Control**  Our combustion simulation framework provides users with a high degree of control over key aspects of the simulation, including ignition location, fire intensity, and airflow, as demonstrated in Fig. 7. Specifically, users can accurately set the ignition point by assigning reaction coordinate variable $Y = 1$ to the target ignition voxel. The perceived fire intensity can be adjusted by increasing the buoyancy force coefficient $\alpha$, which lifts the flames higher, and decreasing the reaction rate $k$, which extends flame visibility—both contributing to a visually stronger fire effect. Airflow can be flexibly controlled by adding an external wind force, enabling users to steer the fire as desired. In addition to these core controls, all other combustion parameters such as thermal diffusivity $\beta$, charring rate $\varepsilon_c$ are also accessible, allowing users to fine-tune the simulation for customized effects.

## 3.3 Combustion Rendering

We introduce the first rendering framework that jointly integrates simulated fire, smoke, and reconstructed 3DGS into a unified volumetric pipeline. It builds upon the reconstructed 3DGS and the grids obtained from Section 3.2, including the reaction coordinate variable $Y$ for fire and smoke and the relative char mass $M_c$ for charring. Using this information, the framework generates the final rendered image that seamlessly combines combustion effects with scene geometry.

Our framework builds upon volumetric rendering (Fong et al., 2017) with targeted simplifications tailored to combustion. Since fire is modeled as a blackbody radiator with negligible scattering, and smoke is treated as a low-albedo medium, we omit scattering terms (Nguyen et al., 2002; Pegoraro & Parker, 2006). The 3DGS is rendered as an opaque background where charring effects are incorporated through $M_c$. Under these assumptions, the radiance $L$ at each pixel is computed as:

$$L = L_{\text{fire}} + L_{\text{smoke}} + \hat{T}(L_{\text{GS}} + L_{\text{phong}}). \tag{3}$$

Here, $L_{\text{fire}}$ and $L_{\text{smoke}}$ are accumulated along the ray before reaching the 3DGS, $\hat{T}$ is the transmittance describing remaining energy, $L_{\text{GS}}$ is the 3DGS radiance with charring, and $L_{\text{phong}}$ models fire illumination on the geometry. The contribution of each term is visualized in Fig. 5. Their computation is given in subsequent rendering passes, with details in Appendix A.3.

**Fire Rendering**  In fire rendering, the dominant visual effect arises from self-emission, which we model based on Planck's blackbody radiation law (Nguyen et al., 2002). The absorption coefficient $\sigma_a$ is set to a fixed positive value when the reaction coordinate $Y > 0$, indicating active combustion, and zero otherwise. Spectral volumetric rendering is then performed by integrating the emission term along the ray, and the resulting spectral distribution is converted to RGB color space with chromatic adaptation, following the approach in (Nguyen et al., 2002), to obtain perceptually plausible colors.

**Smoke Rendering**  Smoke becomes visible as the flame cools down during combustion. We render the smoke when the reaction coordinate variable satisfies $Y \leq Y_{\text{smoke}}$. The smoke color is determined by the type of burning material from material reasoning in Section 3.1. For example, smoke from wood combustion is white, while smoke of burning plastic is black (Fig. 4c). By incorporating this model into the volume rendering pipeline, smoke can be presented along with the fire.

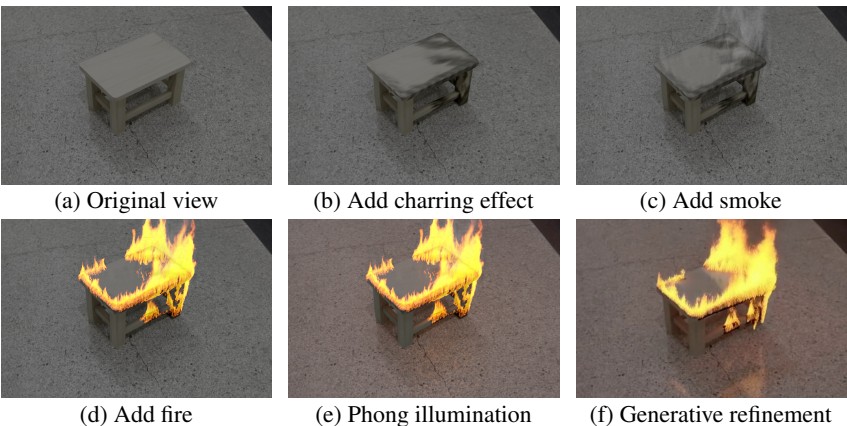

| (a) Original view | (b) Add charring effect | (c) Add smoke |
| (d) Add fire | (e) Phong illumination | (f) Generative refinement |

Figure 5: **Rendering Components Breakdown.** Starting with the original view (a), we first add the charring effect (b). Next, we incorporate the simulated smoke (c), followed by the simulated fire (d). Finally, Phong illumination enhances the ground lighting effect caused by the fire, allowing the originally dark shadow to be brightened (e). An optional generative refinement can further enhance the ground reflection (f).

**3DGS Rendering**  To implement the charring effect in 3DGS, we apply a scaling factor to the color of 3DGS points where the relative char mass satisfies $M_c \geq M_c^{\text{dark}}$. Specifically, the color is dimmed by $r^{\text{dark}} \frac{M_c - M_c^{\text{dark}}}{1 - M_c^{\text{dark}}}$, where $r^{\text{dark}}$ is a user-defined factor (typically less than 1) that controls the degree of color dimming when the char mass reaches its maximum ($M_c = 1$). This approach allows the charred regions to progressively darken as the char mass increases, visually simulating the accumulation of charring on the material surface.

**Phong Illumination**  We adopt the traditional Phong illumination model (Phong, 1998) to simulate the lighting effect of fire on 3DGS. Specifically, we treat voxels with temperatures exceeding a given threshold as volumetric light sources. For each 3D Gaussian, we consider only the diffuse and specular components. The accumulated spectral radiance at each wavelength $\lambda$ is:

$$L_\lambda = \sum_i L_{e,\lambda}^{(i)} \cdot \left[ k_d \left( \mathbf{n} \cdot \mathbf{l}_i \right) + k_s \left( \mathbf{r}_i \cdot \mathbf{v} \right)^s \right],\qquad(4)$$

where $L_{e,\lambda}^{(i)}$ is the spectral radiance emitted by voxel $i$, $\mathbf{n}$ is the surface normal obtained from the normal map rendered by the 3DGS in Section 3.1, $\mathbf{l}_i$ is the light direction, $\mathbf{v}$ is the view direction, and $\mathbf{r}_i$ is the reflection direction. $k_d$ and $k_s$ are the diffuse and specular reflection coefficients, and $s$ controls the sharpness of the specular highlight. Finally, the accumulated spectral radiance is converted into RGB color space using the same way in fire rendering, resulting in the perceived illumination effect on the 3DGS.

**Optional Generative Refinement**  While our method captures key physical aspects of fire, real-world combustion involves additional complexities such as indirect illumination, flickering, and subtle light–material interactions, which remain difficult for physics-based pipelines. To enhance realism, we introduce an optional generative refinement module based on Wan2.1 (Wang et al., 2025), a diffusion video model supporting image and text conditioning. Inspired by SDEdit (Meng et al., 2022) and PhysGen (Liu et al., 2024c), we encode the simulated video into the model's latent space, perturb it with noise, and then denoise it with the first frame as image condition, guided by classifier-free guidance (Dhariwal & Nichol, 2021; Ho & Salimans, 2022). This process adds high-frequency details and more realistic illumination, as shown in Fig. 5f. However, it may also alter background content and lacks strong 3D consistency, so we treat it as an optional refinement step and provide further discussion in Appendix B.6.

## 4  EXPERIMENTS

In this section, we evaluate FieryGS across diverse scenes and compare it with baselines. We further demonstrate the flexible user control of FieryGS. Results highlight FieryGS's strengths in high-fidelity rendering, physical plausibility, and controllable fire synthesis. Please refer to our supplementary video for high-quality dynamic visualizations.

Table 2: Quantitative comparisons.

| Method | Aesthetic Quality↑ | Imaging Quality↑ | DINO Structure↓ |
|---|---|---|---|
| AutoVFX | 0.488 | 0.603 | 1.04 |
| Runway-V2V | 0.605 | 0.701 | 0.68 |
| Instruct-GS2GS | 0.451 | 0.394 | 0.66 |
| **Ours** | **0.624** | **0.702** | **0.38** |

Table 3: User Studies results.

| Baseline | Perceptual Realism | | Physical Plausibility | |
|---|---|---|---|---|
| | Image | Video | Image | Video |
| vs AutoVFX | 88.9 | 77.8 | 86.6 | 85.5 |
| vs Runway-V2V | 79.4 | 66.5 | 85.3 | 79.0 |
| vs Instruct-GS2GS | 85.5 | 63.0 | 83.2 | 84.5 |

*Note:* Values = % of cases where FieryGS is preferred.

**Experimental Details** We evaluate FieryGS on 6 real-world scenes, including 4 custom-captured scenes (*Firewood*, *Kitchen*, *Chair*, *Stool*) recorded with an iPhone, the *Garden* scene from the MipNeRF360 dataset, and the *Playground* scene from the Tanks and Temples dataset. These scenes cover both indoor and outdoor environments and feature diverse object geometries, materials, and spatial arrangements, validating our method in complex, in-the-wild settings.

We compare FieryGS against 3 representative baselines: an automatic VFX pipeline (AutoVFX (Hsu et al., 2024)), a video-to-video generation model (Runway-V2V (Runway, 2024a;b)), and a text-driven 3DGS editing method (Instruct-GS2GS (Vachha & Haque, 2024)). AutoVFX enables dynamic editing in 3DGS scenes via language instructions using Blender's physics engine. Runway-V2V refers to the leading commercial model of Runway for video-to-video synthesis. Instruct-GS2GS performs text-driven editing on 3DGS models via a 2D diffusion model. All support fire synthesis, enabling a comprehensive comparison with our method. All prompts are in Appendix B.1.

**Qualitative Evaluation** Fig. 6 presents a comparison of FieryGS against baselines on *Kitchen* scene, demonstrating dynamic fire synthesis over time. Runway-V2V produces visually appealing fire videos, but significantly alters the original scene's appearance and structure—for instance, a plate originally placed on the table is transformed into a circular groove on the tabletop, and Lego bricks are turned into a pile of wooden blocks. Furthermore, its fire lacks physical plausibility, failing to capture core combustion dynamics such as flame propagation, and it cannot generate smoke colors that vary with different burning materials. AutoVFX incorporates dynamic fire through Blender's physics engine. However, in complex indoor environments, the resulting flames fail to achieve a convincing level of realism. Instruct-GS2GS cannot localize fire edits and supports only static modifications of 3DGS models. In contrast, FieryGS generates temporally coherent fire effects that are both visually authentic and physically grounded, faithfully reproducing the evolution of ignition, flame spread, and scene illumination. More qualitative comparisons are presented in Appendix B.1.

**Quantitative Evaluation** We report **Aesthetic Quality** and **Imaging Quality** scores from VBench (Huang et al., 2024b) to assess visual fidelity, and **DINO Structure Score** (Parmar et al., 2024) to evaluate structure preservation. As shown in Table 2, our method achieves the highest scores in both visual quality metrics and the lowest DINO Structure Score among all baselines, indicating that it produces visually compelling results while faithfully preserving the input scene structure.

**User Studies** We conducted two user studies to evaluate both the perceptual realism and physical plausibility. In the first study (86 participants), users compared 31 randomly sampled image or video pairs and selected the one with more realistic fire that better preserved the background scene. The second study (88 participants) followed the same setup but asked users to judge which result appeared more physically plausible. Results in Table 3 demonstrate a consistent preference for our method. Additional setup details are provided in Appendix B.5.

**Runtime** The average runtime of FieryGS during the simulation and rendering stage is 2.37 seconds per frame on an NVIDIA RTX 4090D GPU. A detailed timing breakdown and comparisons with baselines are provided in Appendix B.4.

**User Control Analysis** A key advantage of FieryGS is its fine-grained user controllability over combustion behavior. Users can adjust the full combustion-related physical parameters—ignition location, airflow, fire intensity, thermal diffusivity, charring rate, and more. Fig. 7 illustrates how varying these parameters produces semantically meaningful and physically consistent changes in fire behavior. For example, altering the ignition location results in different flame propagation paths, while adjusting airflow direction directs the spread of flames accordingly. These controls enable precise authoring of dynamic fire effects without manual 3D modeling or complex simulation setup. Compared to baselines, which either lack explicit control (Runway-V2V), or support only limited, coarse-grained edits (AutoVFX, Instruct-GS2GS), FieryGS offers a significantly more flexible and intuitive editing workflow for physically plausible fire synthesis.

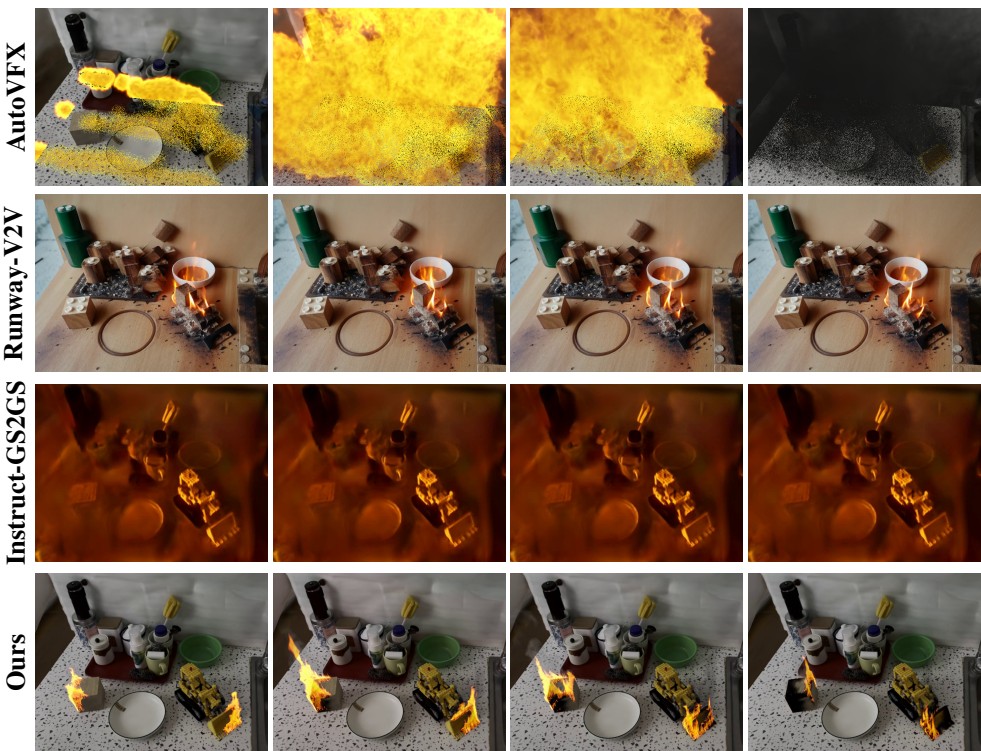

Figure 6: Fire synthesis results over time on *Kitchen* scene. AutoVFX shows limited fire realism in complex indoor environments. Runway-V2V generates visually plausible flames but significantly alters the scene and omits ignition dynamics. Instruct-GS2GS produces static, low-fidelity edits without temporal evolution. In contrast, FieryGS synthesizes physically grounded, time-evolving fire with realistic ignition, spread, and scene illumination.

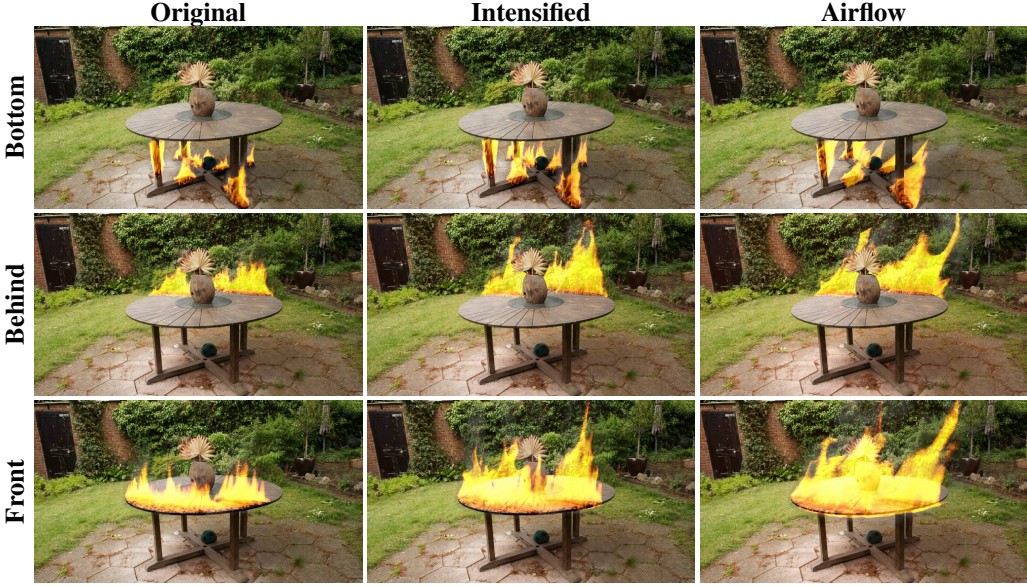

Figure 7: Controllability of FieryGS. Rows vary ignition location: under (Bottom), behind (Behind), and in front of the table (Front). Columns show simulation settings: baseline (Original), increased intensity via stronger buoyancy ($\uparrow \alpha$) and lower reaction rate ($\downarrow k$) (Intensified), and added rightward wind (Airflow). FieryGS enables intuitive control over ignition, intensity, and airflow.

## 5 LIMITATIONS AND CONCLUSIONS

While FieryGS demonstrates strong performance in multi-object scenes, it incorporates several simplifications for efficiency. Specifically, the framework does not explicitly model mass loss or thermal degradation, simplifies certain fire dynamics, and focuses more on multi-object scenes rather than modeling large-scale conflagrations. In addition, the uneven distribution of reconstructed 3DGS points can introduce artifacts, and misclassifications in material reasoning may lead to incorrect combustion behavior. Despite these limitations, FieryGS provides an automated pipeline for in-the-wild fire synthesis, with broad potential for simulation, safety training, and immersive content. Code and data will be released upon acceptance. For a more detailed discussion of limitations and potential directions for future work, we refer readers to Appendix C.

## ACKNOWLEDGEMENT

This work is supported by the projects of Beijing Science and Technology Program(Z251100008125028).

## ETHICS STATEMENT

This work adheres to the ICLR Code of Ethics. We conducted two user studies on Amazon Mechanical Turk to evaluate perceptual realism and physical plausibility. The studies followed platform guidelines, and no personally identifiable information was collected. Beyond these studies, no human subjects or animal experiments were involved. All datasets used in this work were either publicly available or captured in controlled environments, ensuring no violation of privacy or copyright.

One potential societal risk of this research is the misuse of fire synthesis for misinformation or malicious visual manipulation. We explicitly acknowledge this risk and strongly encourage responsible and ethical use. At the same time, we believe that high-quality fire synthesis has significant positive applications. It can benefit a wide range of domains, from AR/VR, gaming, and film production to virtual fire drills, heritage preservation, and robotics perception under adverse conditions, by providing controllable, safe, and realistic fire effects without requiring real-world flame generation, thereby reducing potential risks. We are committed to transparency, integrity, and the responsible dissemination of research outcomes.

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

APPENDIX OVERVIEW

This appendix provides supplementary materials to support and extend the main content of FieryGS. Section A elaborates on implementation specifics for each core component of FieryGS, including scene modeling with combustion property reasoning, combustion simulation, and rendering. Section B presents extended experimental results and analyses, covering additional qualitative comparisons, cost and accuracy analyses of combustion property reasoning, runtime and resource usage, user study setup, and the discussion of optional generative refinement. Section C discusses the current limitations and future directions of our method. We further include a supplementary video, showcasing the dynamic fire synthesis results generated by FieryGS.

## A  METHOD DETAILS

### A.1  SCENE MODELING WITH COMBUSTION PROPERTY REASONING

As outlined in Section 3.1 of the main paper, we first reconstruct a high-quality 3DGS model from multi-view images, accurately capturing both the appearance and geometry of the scene. We then segment the 3D Gaussians and infer combustion-relevant physical properties for each segmented region using a multimodal large language model (MLLM). Below, we provide further implementation details on segmentation and prompt design.

#### A.1.1  HDBSCAN HYPERPARAMETER SETUP

To obtain instance-level 3D segments, we employ HDBSCAN (McInnes et al., 2017) to cluster the feature vectors of 3D Gaussians. We adopt the HDBSCAN parameter settings used in SAGA (Cen et al., 2025), including a minimum cluster size of 10 and an epsilon of 0.01. Inspired by GARField (Kim et al., 2024), we further construct a hierarchy of 3D clusters by recursively applying HDBSCAN at multiple affinity feature scales—specifically 0.9, 0.5, and 0.1. These parameters were selected through empirical validation and remain fixed across all experiments. We found this configuration to generalize well across the diverse scenes in our dataset.

#### A.1.2  PROMPTS FOR COMBUSTION PROPERTY REASONING

A carefully crafted combination of visual and textual prompts is critical to enable accurate material reasoning by the MLLM.

Inspired by previous work (Xu et al., 2024), we design a specialized prompt for GPT-4o tailored to combustion property inference (see Fig. 8). The visual prompt includes a set of contextual images, ranging from global to local perspectives: (1) a full-scene rendering, (2) the same rendering with the target region highlighted using a bounding box and mask overlay, and (3) an isolated and zoomed-in view of the segmented region. This visual hierarchy encourages the MLLM to reason about each part in relation to its global spatial context.

The textual prompt guides the model through a step-by-step reasoning process: it first generates a brief caption describing the segmented region, then selects the most appropriate material type from a predefined material library, and finally infers physical combustion attributes such as burnability and thermal diffusivity. This prompt design enables the MLLM to connect local and global visual cues, and incrementally construct semantic understanding of the scene, facilitating more accurate physical property inference.

### A.2  COMBUSTION SIMULATION

We implement our simulation framework from scratch using the Taichi programming language (Hu et al., 2019), where all variables—including the velocity field $\mathbf{u}$, reaction coordinate $Y$, material temperature $T_m$, and relative char mass $M_c$—are stored at the center of the grid with a resolution of $256 \times 256 \times 256$, following the convention in (Fernando et al., 2004). Based on the operator splitting method (Stam, 2023) for time discretization, the combustion simulation within a single time step $\Delta t$ can be summarized as follows:

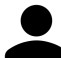

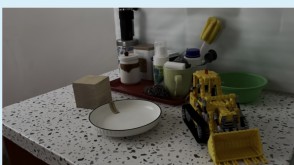 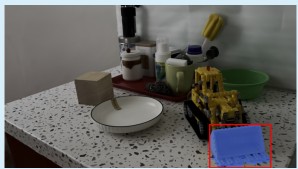 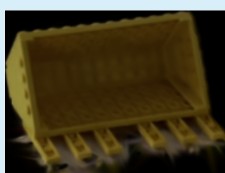

Original Image          Mask Overlay          Part Image

Provided a picture composed of three images arranged from left to right:
1. **Original Image**: The original photo of the entire scene.
2. **Mask Overlay**: A segmentation overlay highlighting the part of interest in blue, with a red bounding box. (This may be either the object or the background.)
3. **Part Image**: A cropped and centered view showing only the segmented part.

You may use all three images to understand what the part is and identify whether the segmented part is an object or the background. The third image (Part Image) provides the clearest visual of the target part, but context from the first and second images may also be useful for identification.

Based on the picture, firstly provide a brief caption of the part.
Secondly, describe what the part is made of (provide the major one).
Thirdly, we combine what the scene is and the material of the part to determine whether the part is burnable.
Finally, you must provide: the thermal diffusivity ratio (i.e., the material's thermal diffusivity divided by that of wood);

Format Requirement:
You must provide your answer as a 4-part tuple:
(caption of the part, material of the part, burnable/unburnable, thermal diffusivity ratio vs. wood)

Do not include any other text in your answer, as it will be parsed by a code script later.

common material library: {wood, sand, metal, plastic, glass, fabric, foam, food, ceramic, paper, leather, plant, stone, cement, concrete, soil, clay, composite}.

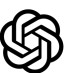

yellow toy dump truck bucket, plastic, burnable, 0.2

Figure 8: Visual and textual prompts used in the MLLM-based combustion property reasoning. The visual input includes both global scene context and a localized rendering of the segmented region. The accompanying text prompt guides the MLLM through a step-by-step reasoning process: it first generates a brief caption describing the segmented region, then selects the most likely material from a predefined material library, and finally infers combustion-relevant physical properties such as burnability and thermal diffusivity.

1. **Advection.** The velocity field $\mathbf{u}$ and the reaction coordinate variable $Y$ are advected using the semi-Lagrangian method (Staniforth & Côté, 1991):

$$\mathbf{u}^* \coloneqq \text{SemiLagrangian}(\mathbf{u}^n, \Delta t, \mathbf{u}^n), \tag{5}$$

$$Y^* \coloneqq \text{SemiLagrangian}(Y^n, \Delta t, \mathbf{u}^n). \tag{6}$$

2. **External Forces and Reaction.** We then account for external forces $\mathbf{f}$ acting on the velocity field $\mathbf{u}$, and for the reaction consumption on $Y$:

$$\mathbf{u}^* \coloneqq \mathbf{u}^* + \mathbf{f}\Delta t, \tag{7}$$

$$Y^{n+1} \coloneqq Y^* - k\Delta t. \tag{8}$$

3. **Pressure Projection.** To enforce the incompressibility condition ($\nabla \cdot \mathbf{u}^{n+1} = 0$), we solve the Poisson equation $\nabla^2 p = \nabla \cdot \mathbf{u}^*$ using Gauss-Seidel iteration to obtain the pressure field $p$. The velocity field is then updated as:

$$p \coloneqq \text{GaussSeidel}(\mathbf{u}^*), \tag{9}$$

$$\mathbf{u}^{n+1} \coloneqq \mathbf{u}^* - \frac{\Delta t}{\rho}\nabla p. \tag{10}$$

For boundary conditions, we apply open boundary condition on the simulation bounding box. For obstacles, open boundary conditions are used when velocity points outward, while no-through (Neumann) boundary conditions are enforced when velocity points inward. This encourages fluid to flow out of obstacles freely but prevents it from entering them.

4. **Charring Effect.** The material temperature $T_m$ and relative char mass $M_c$ are updated explicitly. Since the thermal diffusion term in the update of $T_m$ corresponds to solving a Poisson equation, we subdivide the time step into smaller sub-steps to ensure stability. In this way, our formulation allows us to capture temperature exchange between objects and enables fire propagation between adjacent combustible solids, as illustrated in Fig. 9

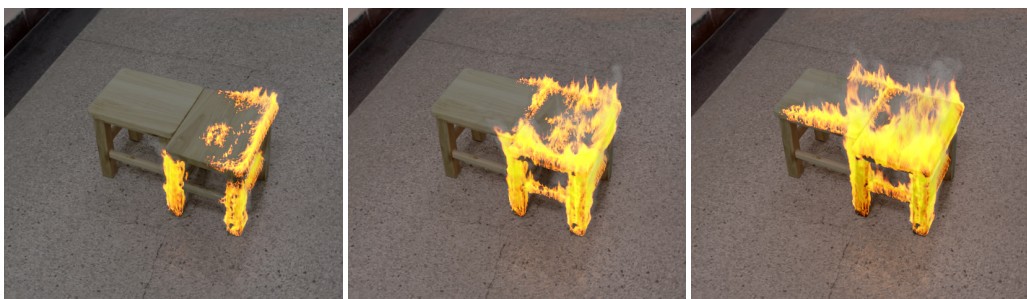

Figure 9: Fire propagation between contacting combustible objects. The three images (left to right) show the gradual spread of fire across different objects. They demonstrate that our model accurately captures thermal diffusion, which enables realistic flame transmission between neighboring flammable materials.

### A.3 Combustion Rendering

To render fire in a physically accurate manner, we first integrate its self-emission spectrum and convert the result into the RGB color space following the approach in (Nguyen et al., 2002; Pegoraro & Parker, 2006). Specifically, the emitted spectral radiance at a given wavelength $\lambda$ is modeled using Planck's blackbody radiation law:

$$L_{e,\lambda}(T) = \frac{2hc^2}{\lambda^5}\frac{1}{e^{\frac{hc}{\lambda kT}} - 1}, \tag{11}$$

where $T$ denotes the local temperature, and $h$, $c$, and $k$ are the Planck constant, the speed of light, and the Boltzmann constant, respectively.

To reduce computational cost, the spectral radiance is first converted to the CIE XYZ color space using the standard tristimulus curves defined by the Commission Internationale de l'Éclairage (CIE), prior

to volume rendering integration (Nguyen et al., 2002; Pegoraro & Parker, 2006). The integrated XYZ values are then transformed into the LMS cone response space using the M_CAT02 transformation matrix. Chromatic adaptation is applied in this space based on the maximum temperature present in the fire (Nguyen et al., 2002). Finally, the result is converted back to the RGB color space, followed by gamma correction for display.

To further enhance the quality of volume rendering, we adopt a coarse-to-fine sampling strategy (Park et al., 2021). We first sample 128 points along each ray uniformly, followed by 1024 points via importance sampling based on the reaction coordinate variable $Y$. Both sets of samples are used for the joint rendering of fire and smoke. To address potential exposure issues when compositing their RGB outputs, we apply ACES tone mapping curve (Narkowicz, 2016) to remap the colors into the $[0, 1]$ range. All these rendering procedures are implemented from scratch in PyTorch (Paszke et al., 2019).

## B    EXTENDED EXPERIMENTAL DETAILS AND RESULTS

### B.1    MORE QUALITATIVE COMPARISONS ACROSS DIVERSE SCENES

**Datasets and Baselines**    We provide additional qualitative comparisons across 6 real-world scenes: 4 custom-captured scenes (*Firewood*, *Kitchen*, *Chair*, *Stool*), the *Garden* scene from MipNeRF360, and the *Playground* scene from Tanks and Temples. For comparison, we consider 3 baselines: AutoVFX (Hsu et al., 2024), a language-driven automatic VFX pipeline; Runway-V2V (Runway, 2024a;b), a commercial video-to-video generation model; and Instruct-GS2GS (Vachha & Haque, 2024), an instruction-based 3DGS editing method.

**Prompt Design**    For Runway-V2V, the prompt is "Add fire to the $\{Target\}$, showing a full burning process — from ignition to full blaze to smoldering ashes. Flames gradually grow, engulf the object, then slowly fade as smoke rises and embers glow."; for AutoVFX and Instruct-GS2GS, we use a shared prompt: "The $\{Target\}$ in the scene is engulfed in roaring flames. The firelight illuminates the surroundings. The smoke billows into the air." In both cases, $\{Target\}$ refers to the manually specified object to be ignited.

**More Qualitative Evaluation**    Beyond the *kitchen* scene comparison shown in the main paper (Fig. 6), we present qualitative results for the remaining 5 scenes in Figs. 12– 16. Runway-V2V generates visually appealing fire effects but significantly alters the rest of the scene—including geometry and appearance of both the background and the burning object—and fails to depict physically plausible combustion dynamics such as ignition, spread, and dissipation. Although AutoVFX is based on Blender's built-in physics engine, it is not specifically designed for fire synthesis and lacks fine-grained control over combustion behavior, resulting in limited visual realism. Instruct-GS2GS performs only coarse, static global edits and is not capable of producing realistic dynamic flames.

In contrast, FieryGS produces photorealistic and physically grounded fire effects that faithfully capture the full progression of combustion, including ignition, flame spread, surface carbonization, and eventual burnout.

### B.2    COST ANALYSIS OF COMBUSTION PROPERTY REASONING

As described in Section 3.1 of the main paper, we employ GPT-4o (Hurst et al., 2024) to perform zero-shot material property reasoning. In our pipeline, the number of API calls corresponds to the number of segmented regions. As summarized in Table 4, FieryGS requires between 9 and 209 calls per scene, depending on scene complexity.

We adopt the ChatGPT-4o-Latest API, which is officially priced at $5 per million input tokens and $15 per million output tokens. On average, each query uses 1,282 input tokens and generates 18 output tokens, resulting in a cost of approximately $0.0066 per call. For a typical scene (mean = 84 calls), the total cost amounts to approximately **$0.55**.

Overall, our GPT-4o-based reasoning pipeline is highly cost-efficient and substantially more economical than manual annotation.

Table 4: GPT-4o API call counts per scene for combustion property reasoning

| Scene | Firewood | Stool | Chair | Kitchen | Garden | Playground | **Avg.** |
|-------|----------|-------|-------|---------|--------|------------|----------|
| Times | 26 | 9 | 46 | 46 | 209 | 169 | **84** |

Table 5: Accuracy of MLLM-based material reasoning across test scenes.

| Scene | Firewood | Stool | Chair | Kitchen | Garden | Playground | **Avg.** |
|-------|----------|-------|-------|---------|--------|------------|----------|
| Accuracy (%) | 88.46 | 88.89 | 82.61 | 91.30 | 89.95 | 89.94 | **89.31** |

### B.3 ACCURACY ANALYSIS OF COMBUSTION PROPERTY REASONING

Accurate and robust combustion property reasoning is essential for physically plausible fire simulation. Here, we quantitatively evaluate the accuracy of our approach.

Since no public scene-level benchmark currently offers reliable ground-truth labels for combustion-relevant materials, we perform a manual evaluation on our 6 test scenes. Specifically, we annotated the material type for each segmented region and compared these annotations against predictions from the MLLM-based material reasoning module. A prediction is deemed correct if it matches the ground-truth label. As summarized in Table 5, our method achieves an average accuracy of **89.31%** across six diverse scenes, demonstrating strong material reasoning capability.

Most material reasoning errors occur in (i) distant background regions or very small objects that are difficult to discern, (ii) heavily occluded areas where the initial segmentation is unreliable, and (iii) occasional reconstruction artifacts in 3DGS that distort texture under the GPT-4o inference view. These limitations are consistent with those of current 3DGS segmentation and vision–language models—limitations shared by current 3DGS segmentation methods and vision–language models. Nevertheless, the overall accuracy is sufficient to support downstream combustion simulation with minimal human intervention.

### B.4 RUNTIME AND COMPUTATIONAL RESOURCES

In our pipeline, 3DGS reconstruction and combustion property reasoning are performed offline, after which pre-frame combustion simulation and rendering are executed. As summarized in Table 6, we report a detailed runtime breakdown of key components, including combustion simulation, Gaussian splatting rendering, and fire and smoke rendering, on a single NVIDIA RTX 4090D GPU. On average, the simulation and rendering stage runs at 2.37 seconds per frame, with peak GPU memory usage below 10.0 GB, demonstrating that our method is both computationally efficient and hardware-friendly.

**Comparison with Baselines** Compared to existing baselines, our method offers a favorable balance of speed and visual quality: AutoVFX Hsu et al. (2024) relies on Blender Blender Online Community (2024) for simulation and rendering and requires approximately 4–10 minutes per frame, making it significantly slower than our method. Instruct-GS2GS Vachha & Haque (2024), which directly edits 3DGS, runs at a fast speed comparable to vanilla 3DGS. However, it produces only coarse, static edits, making it unsuitable for synthesizing realistic dynamic flames. Runway-V2V Runway (2024a;b) is a closed-source model, preventing direct runtime comparisons; according to its official website, generating a 10-second video takes about 30 seconds, but while it produces vivid flame effects, it often alters the background content and lacks both physical plausibility and parameter controllability.

### B.5 USER STUDY SETUP DETAILS

We conduct two user studies on Amazon Mechanical Turk to assess the key aspects of our method: **perceptual realism** and **physical plausibility**. Both studies use an A/B comparison setup, where participants were shown 31 randomly sampled image or video pairs. Each pair included one result

Table 6: Runtime breakdown (s/frame) of key components in FieryGS across different scenes.

| Scene | Firewood | Stool | Chair | Kitchen | Garden | Playground | **Avg.** |
|---|---|---|---|---|---|---|---|
| Simulation | 1.27 | 1.33 | 1.31 | 2.56 | 1.30 | 1.34 | **1.52** |
| GS Render | 0.010 | 0.0045 | 0.0077 | 0.0043 | 0.034 | 0.013 | **0.012** |
| Fire & Smoke Render | 0.75 | 0.90 | 0.86 | 0.69 | 0.45 | 1.37 | **0.84** |

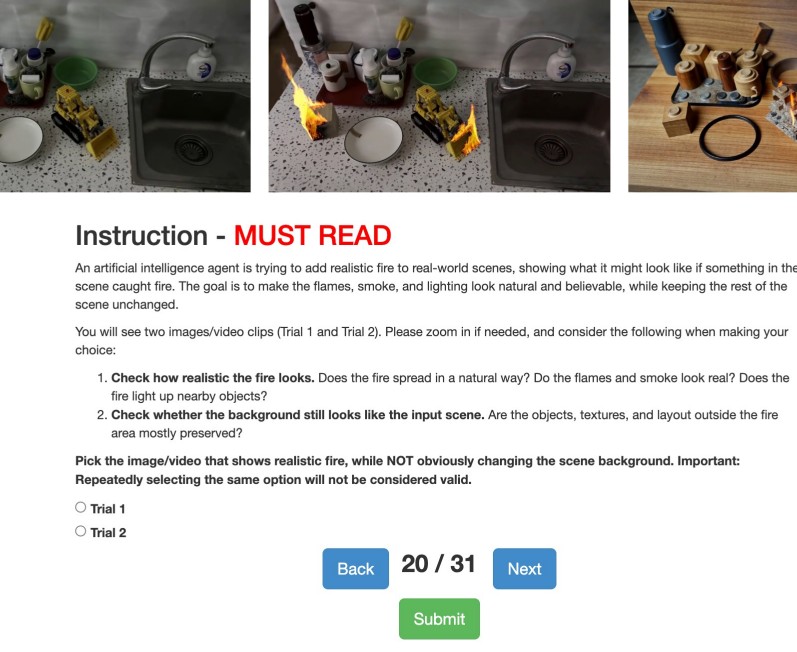

Figure 10: Visualization of interface for user study.

from FieryGS and one from a baseline, with randomized left–right placement to avoid positional bias. An example of the evaluation interface is shown in Fig. 10.

**Study 1: Perceptual Realism**   This study involved 86 participants. In each trial, users were asked to select the result that exhibited more visually realistic fire effects while maintaining the integrity of the original scene. Results are summarized in Table 3 under "Perceptual Realism" (Image/Video).

**Study 2: Physical Plausibility**   We recruited 88 participants using the same evaluation protocol. This time, participants were instructed to choose the version that appeared more physically plausible, based on how consistent the fire behavior was with real-world expectations, while also preserving scene structure. Results are reported in Table 3 under "Physical Plausibility" (Image/Video).

Across both studies, FieryGS consistently outperforms all baselines in user preference for both images and videos. These results indicate that our method produces fire effects that are not only visually compelling but also more aligned with human perception of physical realism.

## B.6   DISCUSSION OF OPTIONAL GENERATIVE REFINEMENT

To balance efficiency and realism, we simplify our combustion simulation and rendering pipeline by omitting certain computationally intensive modules. As a result, our method struggles to capture some high-frequency visual effects, such as complex lighting interactions (e.g., multi-bounce reflections), fine-scale flame textures, and realistic charring patterns.

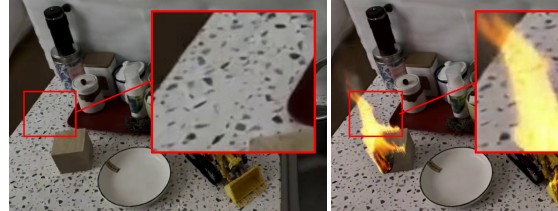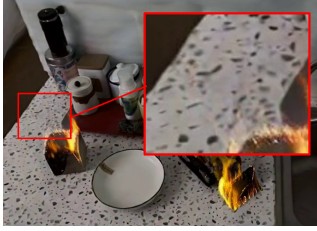

Figure 11: Temporal inconsistency in generative refinement across a fire sequence. While the fire visually improves realism, the underlying table texture—occluded during peak fire—changes after the flame dissipates, revealing the diffusion model's limitations in preserving scene consistency over longer time spans.

To address these limitations, we introduce a video refinement module based on a pre-trained diffusion-based generative model (Wan2.1 (Wang et al., 2025)), as described in Section 3.4. Rather than replacing physics-based simulation, this model is used to complement it—enhancing visual fidelity while preserving physically grounded motion. In practice, we find that this refinement leads to more natural lighting, sharper flame boundaries, and a more compelling overall appearance.

While the refinement model improves visual quality in many aspects, it can introduce two notable side effects that warrant further investigation. First, selectively enhancing fire effects without affecting the background is inherently difficult. Generative models tend to alter surrounding areas along with the target region, and due to the complex and diffuse nature of flame boundaries, masking proves unreliable. Second, as illustrated in Fig. 11, maintaining temporal and 3D consistency remains a challenge, especially for long videos—a limitation rooted in the current capabilities of generative video models themselves.

In summary, while generative refinement opens up new possibilities for achieving photorealistic fire videos, it is still a complementary step that must be carefully integrated with physically-based simulation. We view this as a promising direction for future research, particularly as generative video models continue to evolve in quality and controllability.

## C  DISCUSSION OF LIMITATIONS

Although FieryGS performs effectively on object-level scenes, it also exhibits several limitations that affect both the physical realism of simulated fire and the generality of the framework.

**Material Degradation and Mass Loss**   To maintain efficiency, we do not simulate mass loss or thermal degradation, such as shrinkage, crumpling, or disintegration. Prior works (Liu et al., 2024a; Larboulette et al., 2013) attempt to capture these phenomena but at high computational cost and with complex, manually intensive frameworks. Accurately modeling structural changes with unknown internal material properties (e.g., weakening or collapse) remains extremely challenging (Lakkonen, 2024; Xu & Nan, 2024), representing a significant avenue for future work.

**Simplified Flame and Charring Behavior**   While our model captures turbulent flames and smoke, it simplifies detailed physical processes for efficiency. For example, we do not model how flames ignite surrounding materials, and more physically grounded approaches such as the thin flame model (Nguyen et al., 2002) could better capture dynamic fire behavior. These are important directions for improving the physical fidelity of the simulation.

**Limitations in Scene Scale**   FieryGS is currently tailored to object-level scenes and cannot be directly applied to large-scale scenarios, such as forest or building fires. Extending the framework would require redesigning the fire modeling pipeline and solving new governing equations (Hädrich et al., 2021).

**Non-Uniform 3DGS Distribution**   The reconstructed 3DGS points are unevenly distributed, primarily concentrated on obstacle surfaces, which can introduce artifacts in volumetric simulation and

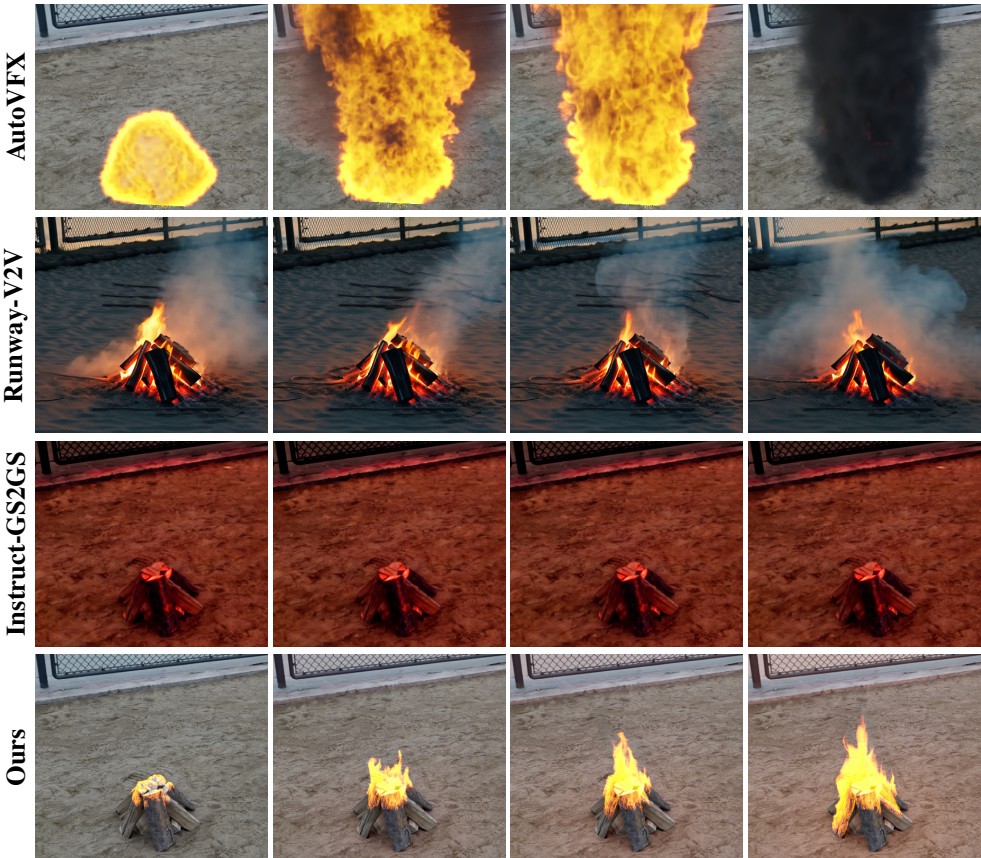

Figure 12: Fire synthesis results over time on *Firewood* scene. AutoVFX produces unrealistic fire and smoke. Runway-V2V generates visually realistic fire, but it completely alters the scene and lacks a gradual ignition process, showing only fully developed flames. Instruct-GS2GS produces static and unrealistic results. In contrast, FieryGS generates realistic, time-evolving fire with a natural ignition and growth process.

rendering. Achieving a more uniform distribution throughout obstacle volumes is therefore another important direction for improvement.

**Misclassifications in Material Reasoning**    As discussed in Section B.3, while FieryGS achieves high accuracy in material property reasoning, misclassifications occur in (i) tiny or distant background objects with limited visual cues, (ii) heavily occluded regions where segmentation quality degrades, and (iii) occasional 3DGS reconstruction artifacts that distort appearance in the GPT-4o inference view; these issues are inherent limitations of current 3DGS segmentation and vision–language models, and future work will focus on improving robustness and reliability under low visibility and occlusion, as well as resilience to reconstruction imperfections.

Despite these limitations, FieryGS provides an automated approach for fire synthesis in complex scenes, enabling applications in simulation, safety, and immersive content creation. Future work will aim to address these constraints to enhance both physical fidelity and scene generalization.

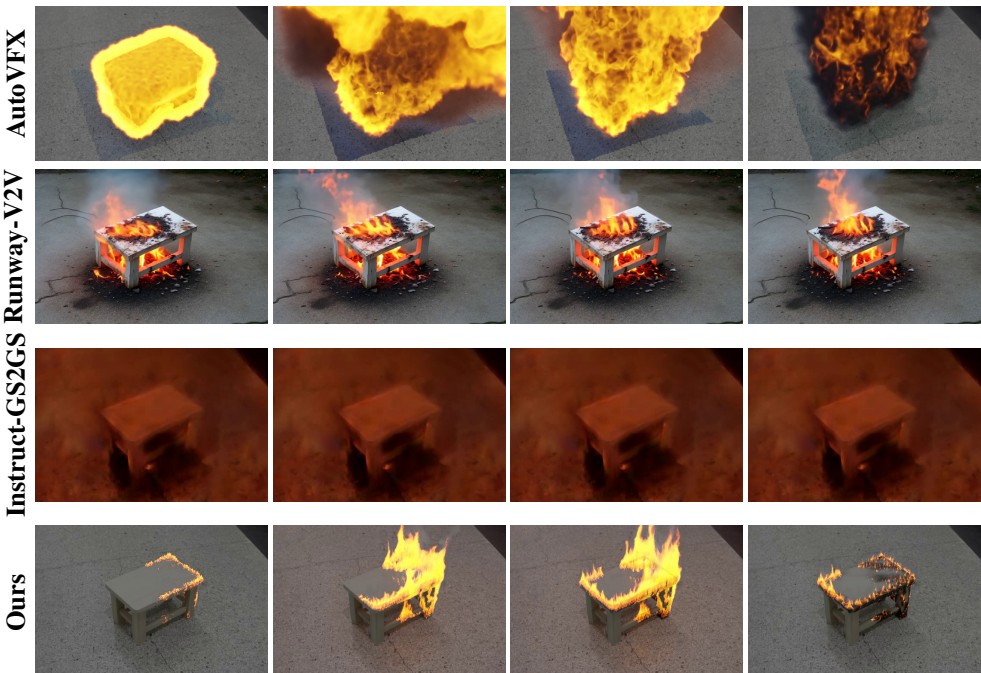

Figure 13: Fire synthesis results over time on *Stool* scene. AutoVFX yields visually implausible results, with exaggerated flames and smoke. Runway-V2V produces realistic-looking fire, but heavily distorts the scene geometry and skips the ignition phase, showing only fully developed flames. Instruct-GS2GS outputs blurry, static edits without dynamic behavior. In contrast, FieryGS produces physically plausible, temporally coherent fire that evolves naturally from ignition to flame spread and decay.

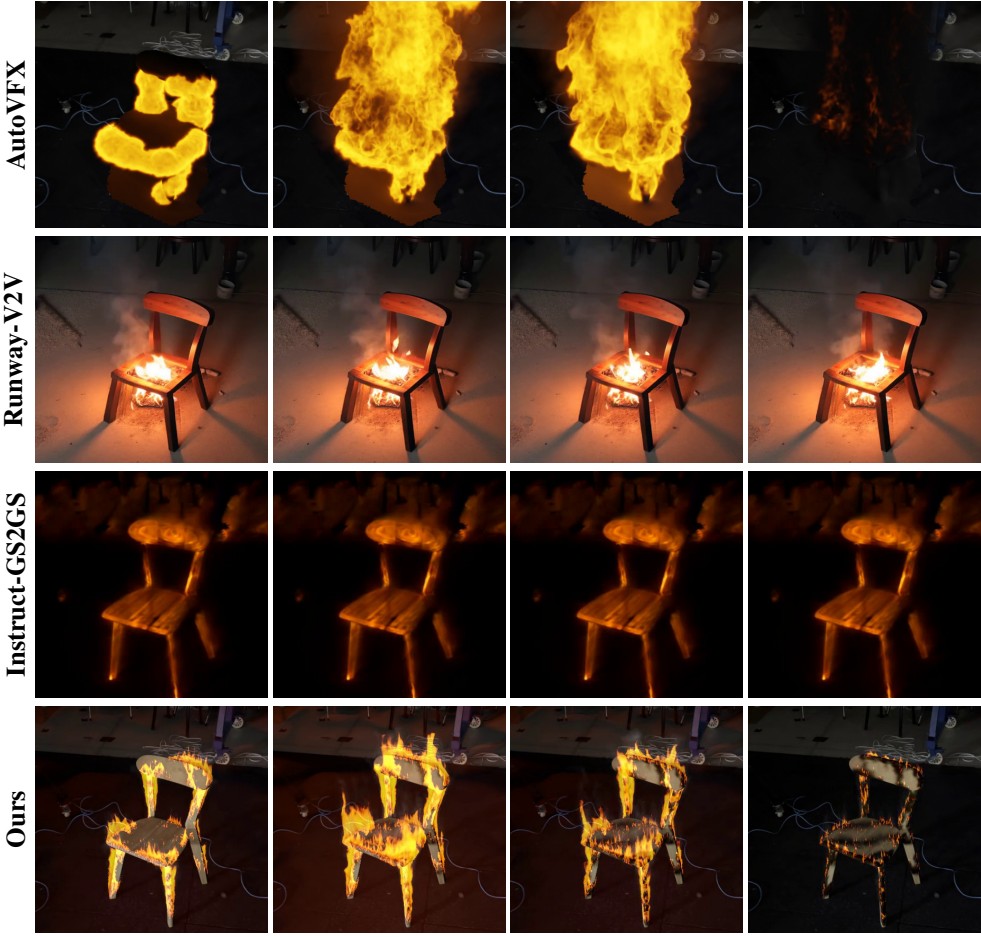

Figure 14: Fire synthesis results over time on *Chair* scene. AutoVFX exhibits exaggerated and implausible fire behavior, with little integration into the scene. Runway-V2V produces visually plausible flames but significantly modifies the scene's appearance and omits the ignition phase. Instruct-GS2GS yields static, glowing effects lacking realistic dynamics. In contrast, FieryGS produces physically grounded fire that evolves naturally—capturing ignition, spread, and burnout—while preserving the underlying scene.

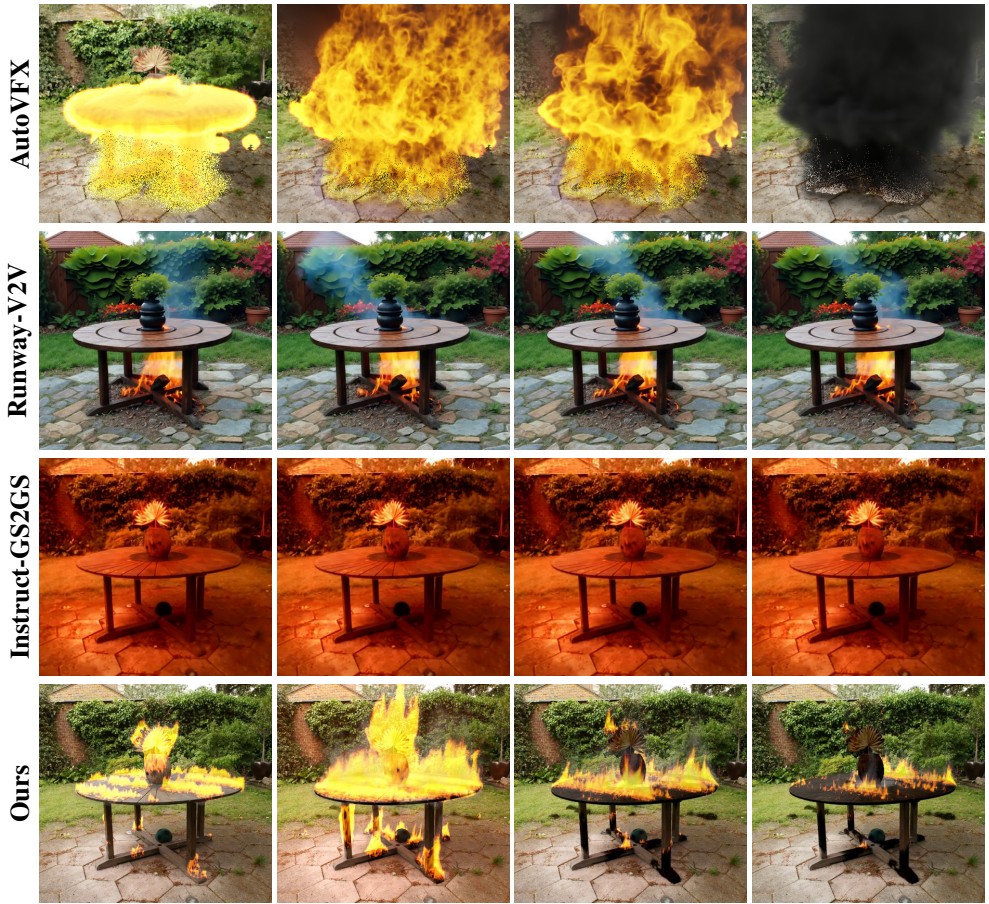

Figure 15: Fire synthesis results over time on *Garden* scene. AutoVFX produces unrealistic, oversized flames and dense smoke that fail to integrate with the environment. Runway-V2V generates visually compelling fire but alters scene details and skips the ignition phase, displaying only intense, fully developed flames. Instruct-GS2GS results in static, overly saturated outputs with no temporal dynamics. In contrast, FieryGS produces physically plausible fire that evolves naturally over time—capturing ignition, spread, and gradual decay—while preserving the original scene context.

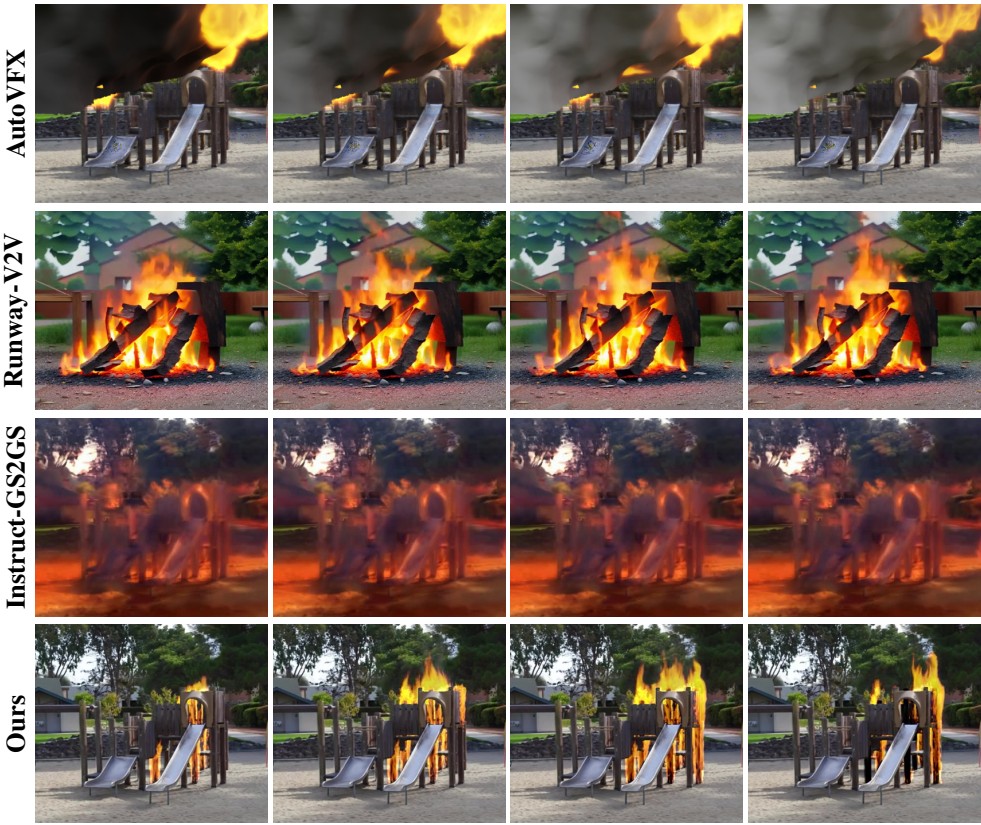

Figure 16: Fire synthesis results over time on *Playground* scene. AutoVFX generates exaggerated fire and dense smoke that appear detached from the physical structure. Runway-V2V produces high-quality flames but drastically alters the geometry and texture of the playground, lacking any notion of progressive ignition. Instruct-GS2GS results in temporally static and visually distorted outputs. In contrast, FieryGS synthesizes physically realistic fire that evolves smoothly over time, preserving scene structure while capturing natural ignition, flame spread, carbonization, and decay.

