# OpenReview forum: "FieryGS: In-the-Wild Fire Synthesis with Physics-Integrated Gaussian Splatting"
_ICLR.cc/2026/Conference — ICLR 2026 Poster_

### Official Review · Reviewer_KF7H · 2025-10-27

**Soundness:** 3
**Presentation:** 3
**Contribution:** 3
**Rating:** 8
**Confidence:** 3

**Summary:**

FieryGS synthesizes realistic, physically grounded fire effects in real 3D scenes by integrating combustion simulation and rendering into the 3D Gaussian Splatting pipeline. It combines scene reconstruction, material reasoning via multimodal LLMs, efficient volumetric combustion simulation,, and unified rendering. The effectiveness of FieryGS is demonstrated by detailed experiments, comprehensive user studies, and visualizations.

**Strengths:**

The proposed method offers a novel, user-friendly, and highly efficient pipeline for realistic fire rendering in complex scenes. By integrating MLLM–based material reasoning with 3DGS–based segmentation, the approach effectively reduces the need for manual parameter tuning and eliminates dependence on traditional expert-level simulation tools. This design democratizes physically-plausible combustion synthesis, allowing users to achieve compelling visual results with minimal technical overhead. The paper is clearly written and well-structured. Extensive quantitative experiments and qualitative visual analyses robustly support the claims.

**Weaknesses:**

I do not have substantial complaints about this paper. I move my concerns to "Questions".

**Questions:**

### Main Question:
- [Comparison to expert-level tools] Since the physical properties can be inferred by MLLMs and the geometry can be modeled by 3DGS, I wonder if all the input required by VFX/CFD (e.g. meshes, material) can also be provided by MLLMs (Line 237). If so, how is the quality and cost comparison between the proposed combustion simulations and expert-level tools? This comparison is more apples-to-apples and can help the readers understand how much the simplified combustion simulation affects the performance.

### Minor Question
- [MLLM comparisons] Is GPT-4o necessary for high-performance? Comparisons with other frontier MLLMs in Table 2/5 and Figure 6 might help readers understand more about how useful the language instructions are.

---

> ### Author Response · Authors · 2025-11-23
> **Response to Reviewer KF7H's comments**
>
> We sincerely thank the reviewer for the constructive suggestions and valuable reviews. We address each comment individually in the responses below:
> - Comparison to Expert-Level Tools (Q1)
> - MLLM Comparisons (Q2)
>
> Please refer to the updated PDF, with all changes highlighted in blue, and to the supplementary videos, provided as individual MP4 files for each new addition.
>
> ---
>
> ### **1. Comparison to Expert-Level Tools (Q1)**
>
> Yes, our 3DGS reconstruction combined with MLLM-based material reasoning provides the inputs typically required by VFX/CFD tools. Here, we evaluate how our framework’s flame simulation compares to expert-level VFX software Blender[1], while emphasizing its unique capability to jointly simulate flame evolution and wood charring. The full experimental setup and visual comparisons have been added in Appendix A.2 (Combustion Simulation Lines 909-959), Figure 11 and the supplementary video (comp_with_blender.mp4) to support this evaluation.
>
> To the best of our knowledge, there is currently no open-source framework capable of fully simulating both flame dynamics and wood charring combustion, making a direct, comprehensive comparison with expert-level tools infeasible. The only system supporting this coupled process is FlameForge[2], which is closed-source.
>
> To provide a partial comparison as requested, we evaluate the flame simulation component of our method against Blender. We provide Blender with the same reconstructed geometry and inferred material properties, run its fire simulation module, and render the output using our renderer for direct visual comparison. This highlights an important point: **while Blender alone cannot synthesize photorealistic and physically plausible combustion for in-the-wild 3D scenes, our full pipeline achieves this**.
>
> **Under identical resolution and ignition settings**, we find that:
>
> - Our simplified simulation produces flame behavior comparable to Blender’s, with only minor deviations in highly turbulent regions,
>
> - Our method is more efficient (**1.27 s/frame** at $256\times 256\times 256$ resolution versus **5.25 s/frame** for Blender).
>
> - Moreover, Blender does not model internal heat conduction or flame spread, leading to stationary flames, whereas our framework naturally captures these processes.
>
> In summary, these results demonstrate that our approach achieves a favorable balance between visual fidelity and computational efficiency, while also enabling the first open-source joint simulation of flame evolution and wood charring in complex 3D scenes.
>
> ---
>
> ### **2. MLLM Comparisons (Q2)**
>
> We appreciate the reviewer's question. Overall, GPT-4o yields the best performance and is therefore our default choice, while other frontier MLLMs can also follow our text–visual instructions and work well within our pipeline.
>
> To evaluate model dependence, we conducted additional experiments with **Qwen3-VL-Plus** using the exact same text-visual instructions. As shown in the table below, Qwen3-VL-Plus achieves a respectable average accuracy of 82.73%, confirming that our text-visual prompting is robust and generalizes well to other frontier MLLMs. However, GPT-4o still provides a clear advantage (89.31%) and shows more reliable material reasoning. Given that GPT-4o is already highly cost-effective (~$0.55 per scene, see Appendix B.2) and offers the highest accuracy for physics simulation, we use it as the optimal choice for our framework, though our pipeline remains flexible to future MLLM advancements.
>
> Regarding downstream metrics (Table 2 and Fig. 6), once the MLLM provides sufficiently accurate material initialization, the visual and physical fidelity is largely determined by the deterministic physics simulation and rendering, rather than by fine-grained differences between MLLMs. Therefore, we believe the material classification accuracy (Table 5) is the most direct metric to assess the impact of different MLLMs. We have updated Appendix B.3 (Lines 1069-1077, Choice of MLLM) and Table 5 in the revised paper to include these comparative results.
>
> | Method | Firewood | Stool | Chair | Kitchen | Garden | Playground | **Avg.** |
> | :--- | :---: | :---: | :---: | :---: | :---: | :---: | :---: |
> | GPT-4o (Default) | 88.46% | 88.89% | 82.61% | 91.30% | 89.95% | 89.94% | 89.31% |
> | Qwen3-VL-Plus | 88.46% | 77.78% | 76.09% | 84.78% | 79.90% | 89.35% | 82.73% |
>
> ---
>
> ### **Summary**
>
> We sincerely thank you for your thoughtful and constructive review. We greatly appreciate the time and effort you dedicated to evaluating our work, and we are glad that you found our pipeline novel, user-friendly, and efficient. We are happy to release code for future research.
>
> ---
>
> ### **Reference**
>
> [1] Blender Online Community: Blender - a 3d modelling and rendering package, blender Foundation, Stichting Blender Foundation, Amsterdam
>
> [2] Liu, Daoming, et al. "FlameForge: Combustion of Generalized Wooden Structures." arXiv preprint arXiv:2412.16735 (2024).

---

> > ### Comment · Reviewer_KF7H · 2025-11-25
> > **Impressive results**
> >
> > Thank the authors for providing the impressive additional results. The additional results have addressed my earlier concerns. I appreciate the overall contribution of the paper and will maintain my rating and confidence score.

---

### Official Review · Reviewer_bvZV · 2025-10-30

**Soundness:** 2
**Presentation:** 3
**Contribution:** 3
**Rating:** 4
**Confidence:** 3

**Summary:**

The paper propose a method to add fire simulation in 3DGS scenes. It first use MLLM to segment flamable objects and then add fire and charring simulation, with good user controllability, including fire intensity, air flow, and color of the fire based on the material.. The final rendering is a combination of flame simulation with phone shading + 3DGS. Experiments show that the proposed method has better visual quality comparing to exisitng baselines. It also proposes an extension with existing generative models to improve photorealism.

**Strengths:**

1. The paper is well-written and easy to understand.
2. The method of using a combination of MLLM and fire/charring simulation is reasonable and easy to implement.
3. The method provides good controllability of the scene setup, esp in controlling the fire color based on the material.

**Weaknesses:**

1. The rendering is one-way. It does not consider the shading of fire on surrounding. Thus causing visual artifacts. Though the generative model may solves the problem in some cases, it may alters the background.
2. The fire simulation does not consider geometry constraint. For example, fire in a building seeing through windows. The simulation of fire / smoke should consider geometry constraints induced by voxels that are "solid".

**Questions:**

1. Is it possible to improve the quality of rendering and fire simulation for the weakness points?
2. What are the aesthetic qualtiy metric? How it is computed?
3. Could u show the visualization of the fire in different control parameters?

---

> ### Author Response · Authors · 2025-11-23
> **Response to Reviewer bvZV's comments**
>
> We sincerely appreciate the reviewer’s thoughtful feedback and constructive suggestions. We address each point in detail below:
> - Improvement on Fire Rendering (W1, Q1)
> - Geometry Constraint for Fire Simulation (W2, Q1)
> - Aesthetic Quality Metric (Q2)
> - Clarification on Visualization of Different Control Parameters on Fire (Q3)
>
> Please refer to the updated PDF, with all changes highlighted in blue, and to the supplementary videos, provided as individual MP4 files for each new addition.
>
> ---
>
> ### **1. Improvement on Fire Rendering (W1, Q1)**
>
> We would like to clarify that our rendering is not one-way. The shading of fire on the surrounding scene is taken into account using Phong illumination, as described in Lines 342–356 of the main paper. In the original submission, diffuse effects tended to dominate, making specular highlights less noticeable. To better demonstrate our rendering capabilities, we re-rendered the Chair scene with more pronounced specular effects (see Appendix A.3 Lines 984-1002, Figure 12 and supplementary video reflection.mp4). Furthermore, as shown in Lines 353-356 and supplementary video (dynamic_lighting.mp4), by introducing temporal fluctuations in light intensity, our method now produces flickering fire effects. We would be happy to consider any further suggestions for enhancing the rendering.
>
> ---
>
> ### **2. Geometry Constraint for Fire Simulation (W2, Q1)**
>
> Our framework explicitly handles geometry constraints in fire simulation. As noted in the paper (Line 215), fire simulation is performed only in air regions, while solid regions are treated as boundaries; the details of boundary handling are provided in the paper (Lines 897-900). This ensures that fire and smoke never penetrate solid voxels. We have also made this more clear by adding description at Lines 215-237.
>
> To further illustrate this capability, we conducted an additional experiment in the Firewood scene by placing a virtual brick above the campfire, which can be found at Appendix A.2 Lines 900-902, Figure 9 and supplementary video (add_brick.mp4). Although this is a simplified setup, the experiment clearly shows that the fire splits into two streams when the brick is present, demonstrating behavior completely different from the original simulation and confirming that solid-voxel geometry constraints are respected.
>
> ---
>
> ### **3. Aesthetic Quality Metric (Q2)**
>
> Thank you for pointing this out. As stated in the paper (Line 405), we adopt the Aesthetic Quality metric from VBench [1], a widely used benchmark evaluating image and video quality. As described in VBench, this score is computed using the LAION Improved Aesthetic Predictor, which consists of: a CLIP ViT-L/14 image encoder pretrained on large-scale image-text data, followed by a lightweight linear regression head trained on the AVA (Aesthetic Visual Analysis) dataset, where each image is annotated with human-rated aesthetic scores on a 1–10 scale. To evaluate a generated video, we extract all frames, compute the aesthetic score for every frame using this predictor, and report the average score across all frames as the final Aesthetic Quality metric. This approach provides a standard, efficient, and human-aligned proxy for subjective visual aesthetics.
>
> ---
>
> ### **4. Clarification on Visualization of Different Control Parameters on Fire (Q3)**
>
> We would like to clarify that we have included a comprehensive visualization of these effects in Lines 420-424 and Figure 7 of the main paper, as well as the original supplementary video (02:56-03:12), which demonstrates the controllability of FieryGS in three key aspects: (1) Ignition Location, where we show fire originating from different points (e.g., bottom, behind, or front of the table); (2) Fire Intensity, where we visualize fire under different “intensity” parameters (adjusted buoyancy force $\alpha$ and reaction rate $k$); and (3) Airflow, where we visualize the effect of external wind forces directing the flame spread.
> Beyond these three factors, FieryGS also supports additional controllable parameters that further modulate the fire behavior (see Line 420).
>
> ---
>
> ### **Summary**
>
> We sincerely thank you for the thoughtful and constructive feedback. We have carefully addressed all the concerns raised. We believe our approach provides a valuable and practical solution for automated and controllable volumetric fire synthesis, and we kindly hope you would consider revisiting your score. We remain open to any further discussion.
>
> ---
>
> ### **Reference**
>
> [1] Huang, Ziqi, et al. "Vbench: Comprehensive benchmark suite for video generative models." Proceedings of the IEEE/CVF Conference on Computer Vision and Pattern Recognition. 2024.

---

### Official Review · Reviewer_FTxR · 2025-11-01

**Soundness:** 4
**Presentation:** 3
**Contribution:** 2
**Rating:** 6
**Confidence:** 3

**Summary:**

The paper present FireyGS, which is a framework that physically simulates fires and charring of objects in real-world images. The process is fully automated with some user control.

This is achieved through:
1) making volumetric representation of the real-world scene using 3D Gaussian Splatting,
2) letting a multi-modal LLM reason about the material characteristics of objects, such as whether they are consumable by fire,
3) physically simulate fire and smoke using the Navier-Stokes equations while also modelling charring, and
4) rendering the fire using Phong Illumination and an optional generative model.

This allows to easily integrate fires into real-world scenes with some user control, such as adjusting fire intensity and airflow, which is quite impressive. The rendering time is very short. The evaluation shows that state-of-the-art methods are far from achieving this.

**Strengths:**

* The paper is well written and structured well with many images illustrating the capabilities of the method.
* The results are quite impressive. The paper illustrates how fire can be put into realistic images with little effort.

The fact that one “easily” can add simulated fires to real-world images is quite astonishing to me. I am very impressed by the results.

**Weaknesses:**

* The novelty is limited to using an LLM for reasoning about material characteristics and putting well known methods together. If I get this right, there is nothing new about the 3D Gaussian Splatting method, the fire and charring simulation, nor the illumination method.
* While the results look good, they are far from realistic, especially for larger, more turbulent fires. This can easily be seen by comparing Figure 2 (real-world, live fire) and the images generated by the framework. The smoke and turbulence in the real-world images makes for deeper and visually more “exciting” flames. This is probably because the low resolution of the fire, also the turbulence is not modelled well by the simplified Navier-Stokes. This method works fairly well for small fires with little turbulence – in my opinion. The graphics community, however, have many methods and tricks that would help here, such as dynamic meshes that allows detail to be increased in the areas where there is turbulence. Other more complex and realistic physical simulations have been proposed as well. This would if course hamper computational performance, which is really good for the method presented here. The good thing though is that there are many ways to improve it based on related work in physics-based animation of fire.
* Some parts could be better explained. It is not clear to me what exactly is fed to GPT4o to reason about material characteristics. I would assume the real-world images, but it is not clear to me from the text.

On one hand, while the results are impressive, I wonder if the novelty falls short. This paper represents a fairly small step in an evolution, not a revolution. On the other hand, one could easily see that the significance of this work could be high, as it shows how fire could be simulated in real-world scenes.

While the rendering of the fire leaves much to be wanted (think about the vivid fires and explosions in special effects such the opening scene of Star Wars III: Revenge of the Sith – from 2005 where explosions also are simulated using physics), this does not affect my recommendation much, as I expect this to be improved quickly by future work.

Because of this I lean towards acceptance.

**Questions:**

* On line 126, it is implied that the method presented in the paper “maintains physical accuracy”. Is maintaining physical accuracy a good term? The reason I ask is that I would assume that this would be evaluated in some way. It is not, and it is not easy to do either as far as I am concerned.
* The charring effect is very nice. Why would you not let the fire consume the fuel (material) in each voxel? Have you tried? In theory, this should be straight forward. Are there issues with the visualization?
* It is stated on line 215: “Focusing on efficiency, our method simplifies processes …”. Could you please be explicit about which processes? I would like to understand exactly what you mean.
* Figure 5f: While the generative refinement improves illumination shadows, it seems to me that the flames get bleaker with less details. Is this true and a problem in general, or is it just this image? What is going on here - if it is the case? Please help me understand.
* Line 409: “A detailed timing breakdown and comparisons with baselines are provided …”. It would be nice to see the running times for the baselines as well.

---

> ### Author Response · Authors · 2025-11-23
> **(1/3) Response to Reviewer FTxR's comments**
>
> We are grateful for the reviewer’s valuable comments and insightful suggestions. Our detailed responses to each concern are as follows:
> - Clarification on Novelty (W1)
> - Clarification on Simplified Simulation and Future Direction (W2, Q3)
> - Clarification on GPT-4o Input (W3)
> - Clarification of “Physical Accuracy” (Q1)
> - Voxel Consumption (Q2)
> - Clarification on Generative Refinement (Q4)
> - Running Time for Baselines (Q5)
>
> Please refer to the updated PDF, with all changes highlighted in blue, and to the supplementary videos, provided as individual MP4 files for each new addition.
>
> ---
>
> ### **1. Clarification on Novelty (W1)**
> Thank you for raising this question. We would like to clarify that the novelty of FieryGS extends beyond simply combining existing tools. It lies in the **tight integration** and the **specific technical adaptations** required to make physically grounded fire synthesis possible in in-the-wild 3DGS scenes—a capability that did not previously exist.
>
> 1. **The First Holistic Framework for In-the-Wild Fire Synthesis**. As the reviewer noted, our work "shows how fire could be simulated in real-world scenes." Prior to FieryGS, achieving this required either expensive manual geometry modeling (CFD/VFX) or relied on uncontrollable video generation lacking physical consistency. We propose the **first framework** that automates this entire pipeline: from capturing a real scene to inferring its physical properties, simulating dynamics, and rendering photorealistic results. This system-level innovation enables applications (e.g., virtual fire drills in a scanned real-world scene) that were previously infeasible.
>
> 2. **Tailored Technical Adaptations for an Efficient Integrated Pipeline**. Simply combining existing methods is insufficient. We introduce several tailored designs:
>
> - **Zero-Shot MLLM-based Material Reasoning**: We present a tailored pipeline that lifts the 2D material inference capability of MLLMs to combustion-relevant materials in in-the-wild 3DGS scenes. We segment Gaussians into coherent 3D regions, render each region from its most informative viewpoint, query an MLLM with a carefully crafted visual-text prompt, and back-project its predictions to obtain a consistent, combustion-aware 3D material field for simulation initialization, **without any manual material annotations**.
>
> - **Efficient Volumetric Combustion Simulation**: We present the first open-source framework that **jointly simulates flame dynamics and wood-charring combustion**. By simplifying physical models to strike a balance between visual realism and computational cost, our method runs efficiently—achieving 1.27 s/frame at a $256\times 256\times 256$ resolution, compared to 5.25 s/frame for fire-only simulation in Blender. The detailed comparison settings can be found in our newly added Appendix A.2 (Lines 909-959), Figure 11 and the supplementary video (comp_with_blender.mp4).
>
>
> - **Unified Volumetric Rendering for 3DGS**: Integrating fire rendering with 3DGS is non-trivial. We propose the first unified volumetric rendering framework that seamlessly combines simulated fire, smoke, and 3DGS with charring and fire illumination. This handles the depth compositing and occlusion between the explicit geometry and volumetric effects correctly, which is not a standard feature of 3DGS.
>
> In summary, FieryGS is not merely a combination of existing tools, but a holistic system that overcomes specific interdisciplinary challenges to enable a new capability: automatic, visually realistic, and physically grounded fire synthesis in the wild.

---

> > ### Author Response · Authors · 2025-11-23
> > **(2/3) Response to Reviewer FTxR's comments**
> >
> > ### **(Continue)**
> > ---
> >
> > ### **2. Clarification on Simplified Simulation and Future Direction (W2, Q3)**
> >
> > Thank you for the thoughtful and constructive feedback. In designing our simulator, we intentionally adopt a trade-off between physical accuracy and computational efficiency, which allows us to achieve efficient simulation—**1.52 s/frame** on average at a resolution of $256\times 256\times 256$, as shown in Table 6. As a consequence of this design choice, and as you correctly pointed out, while our results remain visually plausible for small-scale fires, they are indeed less realistic for larger and more turbulent flames.
> >
> > To achieve an efficient framework that can jointly simulate flame evolution and wood charring, **we adopted several simplifications in both components**.
> >
> > **For flame simulation**:
> >
> > - We use an incompressible model rather than a compressible formulation (Lines 256–258);
> > - We approximate temperature using a quadratic function of the reaction coordinate Y instead of solving the full temperature PDE (Lines 260–263);
> > - We apply Gauss–Seidel iterations for pressure projection for efficiency (Lines 890-896).
> >
> > **For wood charring**:
> >
> > - Voxels undergoing combustion are assigned a fixed burn temperature T_burn​ (Lines 269–270);
> > - We use a simplified single-stage charring model without explicitly modeling insulation-layer formation or volatile release (Lines 274–275).
> > We have also clarified the statement on Lines 238-241 of the main paper to make the description of these simplifications more explicit.
> >
> > These choices make the simulator lightweight and efficient, but they naturally impose limitations on the realism of the combustion results. As you correctly point out, incorporating more advanced techniques—such as adaptive meshing, turbulence-enhancing schemes, or more complete physical formulations—could further improve fidelity, albeit at the cost of increased computational expense. We view the integration of such methods as an exciting avenue for future work. Similarly, we acknowledge that rendering quality can be further enhanced, and we leave this as a promising direction for future exploration.
> >
> > ---
> >
> > ### **3. Clarification on GPT-4o Input (W3)**
> >
> > Thank you for pointing this out. To clarify, the inputs to GPT-4o are 2D renderings from the reconstructed 3DGS model, alongside a tailored textual prompt, which can be found in Section 3.1 (MLLM-based Combustion Property Reasoning) and detailed in Appendix A.1.2 / Figure 8.
> >
> > Specifically, for each segmented 3DGS region, we first calculate the viewpoint where the target region has the highest visibility. From this viewpoint, we generate **a three-panel RGB composite consisting of: (1) the full scene rendering; (2) the same rendering with the target region highlighted by a bounding box and mask overlay; and (3) an isolated, zoomed-in view of the segmented region**. This composite, combined with **a structured textual prompt** that guides the reasoning process, serves as the input to GPT-4o. We have revised Section 3.1 (Lines 199-204), Appendix A.1.2 (Lines 791), and the caption of Figure 8 to explicitly clarify this.
> >
> > ---
> >
> > ### **4. Clarification of “Physical Accuracy” (Q1)**
> >
> > Thank you for the comment. We agree that “maintains physical accuracy” is too strong and not appropriate here. We have revised the wording to “maintains physical plausibility” on Line 126 of the main paper, which better reflects our intent.
> >
> > ---
> >
> > ### **5. Voxel Consumption (Q2)**
> >
> > Thank you for the question. We have experimented with allowing the fire to consume material in the voxel, and Fig. 16 in the Appendix shows an additional experiment in which the flame consumes the lower crossbeam of the chair. However, this approach introduces several challenges:
> >
> > - Exposure of reconstruction artifacts. Our geometry is reconstructed using 3DGS, which provides high-quality surface representations but does not capture meaningful interior volume. Once the burned surface voxels are removed, the newly exposed interior typically contains severe artifacts, resulting in visually implausible outcomes.
> >
> > - Lighting and shadow effects. When voxels are removed, modeling the resulting changes in shadows and illumination accurately is difficult with our current renderer.
> >
> > For these reasons, although voxel consumption is physically intuitive and straightforward to simulate, we chose not to include it in the main pipeline. We have added a discussion of this point in Appendix C (Lines 1242-1257, Material Degradation and Mass Loss).

---

> > > ### Author Response · Authors · 2025-11-23
> > > **(3/3) Response to Reviewer FTxR's comments**
> > >
> > > ### **(Continue)**
> > > ---
> > >
> > > ### **6. Clarification on Generative Refinement (Q4)**
> > >
> > > Thank you for raising this point. The reduction in fine flame details is a general characteristic of the diffusion-based generative refinement module.
> > >
> > > As described in Sec.3.3 (Optional Generative Refinement), this module employs a video diffusion model (Wan2.1) via an SDEdit-style process to enhance photorealism. By encoding the simulated video into the latent space, adding noise, and then denoising it, the model effectively synthesizes more realistic global illumination (e.g., shadows and reflections). However, this process inherently resynthesizes the fire texture based on the model's learned priors. This tends to replace specific physical details with a more "generalized" fire appearance. Consequently, high-frequency details of the simulated flames may be smoothed out, or their brightness altered ("bleaker"), to harmonize them with the background ambient light.
> > >
> > > This trade-off is precisely why we designate this module as optional. Our core framework prioritizes physical fidelity and controllability, while the refinement is offered as a post-processing step for scenarios where photorealistic lighting integration is the primary goal.
> > >
> > > ---
> > >
> > > ### **7. Running Time for Baselines (Q5)**
> > >
> > > We would like to clarify that we have included the runtimes for baselines in Appendix B.4(Lines 1131-1146, Comparison with Baselines), which is referenced in Section 4 (Line 417) of the main paper.
> > >
> > > In our experiments, (1) AutoVFX relies on Blender’s physics engine and is significantly slower, requiring approximately 4–10 minutes per frame; (2) Runway-V2V, as a commercial closed-source model, takes about 30 seconds to generate a 10-second video (approx. 0.125s/frame at 24fps), but lacks physical control and consistency; and (3) Instruct-GS2GS runs at a speed comparable to vanilla 3DGS (real-time) but is limited to static edits. In contrast, our method achieves 2.37 seconds per frame, offering a favorable balance between high-fidelity physical simulation and computational efficiency.
> > >
> > > ---
> > >
> > > ### **Summary**
> > >
> > > We sincerely thank you for your thoughtful and constructive review. We greatly appreciate the time and effort you dedicated to evaluating our work, and we are pleased that you found our approach for adding simulated fires to real-world scenes useful. We are also happy to release our code to facilitate future research.

---

### Official Review · Reviewer_EZSh · 2025-11-03

**Soundness:** 3
**Presentation:** 3
**Contribution:** 2
**Rating:** 6
**Confidence:** 3

**Summary:**

This work presents a system named “Physics-Integrated Gaussian Splatting,” aiming to synthesize
controllable and physically inspired fire and combustion effects in real-world 3D scenes.  The method integrates several coordinated modules to form a complete pipeline from real-scene  reconstruction to fire simulation:
 1) a multimodal large model predicts material types and combustion-related parameters in 2D projection
space and back-project them to 3D Gaussian
 2) the 3D scene is voxelized, where solid and air regions are distinguished by Gaussian density—simplified
 fire simulations are applied to air voxels, while thermal diffusion and charring are modeled for solids, with
intuitive user controls such as ignition position and airflow
 3) a unified rendering framework generates multiple visual effects, including flames, smoke, charring, and
indirect illumination.

The work does not introduce a new rendering theory or physical model but integrates existing techniques
 3D Gaussian Splatting, simplified fluid simulation, and multimodal reasoning—into a complete interactive
 pipeline, producing visually realistic fire results.

 Experiments on multiple real and synthetic scenes demonstrate visually realistic and controllable fire
generation, and the results appear to outperform existing generation-based approaches

**Strengths:**

This paper presents an integrated system combining 3D Gaussian Splatting, simplified physical simulation, and multimodal reasoning to construct a physically-informed and controllable fire generation pipeline.

This integration and simplification provide a certain degree of novelty in application, making complex fire simulation more practical and easier to operate. The system is well-designed, with modules for material prediction, fire simulation, and rendering fire results. Experiments on multiple real and synthetic scenes demonstrate that the generated fire is visually plausible and controllable, outperforming existing generation-based methods. The user interaction design further enhances the system’s operability.

The paper provides clear and understandable descriptions of the system modules  and workflow, and the authors indicate that the source code will be released in the future, which facilitates follow-up
research. Overall, this work offers a complete and practical solution for interactive fire generation, with utility for computer graphics, visual effects, and virtual environment fire simulation

**Weaknesses:**

While the integration offers practical value, the method has limited theoretical and technical novelty.  Moreover, the approach heavily depends on the MLLM’s 2D material inference capability, and its performance on uncommon materials, composite materials, or extreme fire conditions remains unexplored.
Additionally, the paper does not provide comparisons between the simplified physical simulation and a full
physics-based simulation, making it difficult to justify the acceptability of the simplifications in practice. Finally
 While FieryGS accounts for multiple effects of fire combustion, it does not explicitly capture or evaluate the
dynamic lighting effects generated by fire, which contribute to the perceived motion and liveliness of the
scen

**Questions:**

- How does the system perform on uncommon materials, composite materials, or objects with unusual textures?

- Could the authors clarify or quantify how much these physical simplifications affect the realism of the generated fire?

---

> ### Author Response · Authors · 2025-11-23
> **(1/2) Response to Reviewer EZSh's comments**
>
> We appreciate the reviewer’s insightful comments and constructive feedback. Below, we provide detailed responses to each concern as follows:
> - Clarification on Novelty (W1)
> - MLLM’s Performance on Challenging Materials (W2, Q1)
> - Comparison with Full Physics-based Simulation (W3, Q2)
> - Dynamic Lighting Effects (W4)
>
> Please refer to the updated PDF, with all changes highlighted in blue, and to the supplementary videos, provided as individual MP4 files for each new addition.
>
> ---
>
> ### **1. Clarification on Novelty (W1)**
>
> Thank you for raising this point. Our contribution lies in several tailored designs that enable the system to work seamlessly together, making physically grounded fire synthesis possible in in-the-wild 3DGS scenes—a capability that was not previously available.
>
> - **Zero-Shot MLLM-based Material Reasoning**: We present a tailored pipeline that lifts the 2D material inference capability of MLLMs to combustion-relevant materials in in-the-wild 3DGS scenes. We segment Gaussians into coherent 3D regions, render each region from its most informative viewpoint, query an MLLM with a carefully crafted visual-text prompt, and back-project its predictions to obtain a consistent, combustion-aware 3D material field for simulation initialization, **without any manual material annotations**.
>
> - **Efficient Volumetric Combustion Simulation**: We present the first open-source framework that **jointly simulates flame dynamics and wood-charring combustion**. By simplifying physical models to strike a balance between visual realism and computational cost, our method runs efficiently—achieving 1.27 s/frame at a $256\times 256\times 256$ resolution, compared to 5.25 s/frame for the fire-only simulator in Blender.
>
> - **Unified Volumetric Rendering for 3DGS**: Integrating the rendering of simulated fire and smoke with 3DGS is non-trivial. We propose the first unified volumetric rendering framework that seamlessly combines simulated fire, smoke, 3DGS with charring and fire illumination. This correctly handles depth compositing and occlusion between explicit geometry and volumetric effects, which is not supported by standard 3DGS renderers.
>
> In summary, FieryGS is not a simple combination of existing tools; it is a **unified system with targeted technical contributions** that bridge 3DGS, MLLM-based material reasoning, and volumetric combustion simulation and rendering, enabling automatic, visually realistic, and physically grounded fire synthesis in in-the-wild scenes.
>
> ---
>
> ### **2. MLLM’s Performance on Challenging Materials (W2, Q1)**
>
> Thank you for pointing this out. To validate the robustness of our MLLM-based material reasoning on these challenging cases, we conducted additional qualitative experiments on diverse real-world images. The results are included in Appendix B.3 (Lines 1079-1120, MLLM Robustness on Challenging Materials) and Figure 13 in the revised manuscript:
>
> - **Uncommon/Composite Materials**: For objects made of mixed or rare materials, we explicitly includes a "composite" category. As shown in Figure 13(a), for a car exhaust diffuser (a complex part often made of carbon fiber or hard plastics), the model correctly classified it as "composite" and inferred it was "unburnable" based on the semantic understanding of the object's function and context.
>
> - **Unusual Textures**: The model successfully looks beyond surface patterns to infer underlying material properties. As shown in Figure 13(b), when tested on barbed wire (with dense, repetitive geometric textures), GPT-4o correctly identified the underlying material as "metal" (unburnable), effectively prioritizing physical semantics over visual appearance.
>
> These examples suggest that the MLLM exhibits strong inference capabilities for challenging materials, providing a sufficiently robust basis for downstream physical simulation and rendering.

---

> ### Author Response · Authors · 2025-11-23
> **(2/2) Response to Reviewer EZSh's comments**
>
> ### **(Continue)**
> ---
>
> ### **3. Comparison with Full Physics-based Simulation (W3, Q2)**
>
> Thank you for raising this question. We have added a comparison experiment to directly assess how much our physical simplifications influence visual realism. To better support this conclusion, we have also added the full experimental setup and visual comparisons in Appendix A.2 (Lines 909-959), Figure 11 and the supplementary video (comp_with_blender.mp4).
>
> We fully agree that comparing our simplified combustion model against a more complete physical simulation would be ideal. However, to the best of our knowledge, **no open-source framework currently supports joint simulation of flame dynamics and wood-charring combustion**. The only system capable of this coupled process is FlameForge [1], which is closed-source and thus unavailable for benchmarking.
>
> To partially address this concern, we compare the flame-simulation component of our method with Blender [2]. To enable a fair comparison, we feed Blender with the same input used by our simulator—namely, the occupancy grid obtained from our 3DGS reconstruction combined with MLLM-based material reasoning. Blender then runs its fire-simulation module on this grid, and we render its output using our renderer for direct visual comparison. This also highlights an important aspect of our contribution: **Blender alone cannot complete the task of synthesizing photorealistic and physically plausible combustion effects for in-the-wild 3D scenes**, whereas our full pipeline makes it technically feasible.
>
> **Under identical resolution and ignition settings**, we find that:
>
> - **Our simplified simulation produces flame behavior comparable to Blender’s results, with only minor deviations for more turbulent flames but remaining within an acceptable accuracy range**.
>
> - Meanwhile, our method is more efficient—**1.27 s/frame** at a resolution of $256\times 256\times 256$ compared with **5.25 s/frame** for Blender.
>
> - Furthermore, Blender does not model heat conduction inside the wood or the resulting flame spread, leading to a stationary flame. In contrast, our framework **naturally captures internal heat transfer and flame propagation**.
>
> In summary, although a full multiphysics comparison is not currently possible, our additional experiment demonstrates that the simplifications in our simulator have a limited effect on realism while achieving higher efficiency and supporting physical processes not available in existing open-source tools.
>
> ---
>
> ### **4. Dynamic Lighting Effects (W4)**
>
> Thank you for pointing this out. We agree that incorporating a flickering effect can further enhance the realism of fire illumination. To more comprehensively demonstrate this aspect, we extended our lighting model to include temporal fluctuations in emitted light intensity using Perlin noise. We also added an additional experiment on the Firewood scene with reduced background brightness, making the resulting dynamic lighting clearly observable. A detailed description of this implementation has been added in the main paper (Lines 353-356), and the corresponding experimental results are included in the supplementary video (dynamic_lighting.mp4).
>
> ---
>
> ### **Summary**
>
> We thank the reviewer once again for the thoughtful feedback and the time dedicated to reviewing our work. We are happy to provide further clarifications in additional discussions. Thank you!
>
> ---
>
> ### **Reference**
>
> [1] Liu, Daoming, et al. "FlameForge: Combustion of Generalized Wooden Structures." arXiv preprint arXiv:2412.16735 (2024).
>
> [2] Blender Online Community: Blender - a 3d modelling and rendering package, blender Foundation, Stichting Blender Foundation, Amsterdam

---

### Author Response · Authors · 2025-12-03
**Paper Rebuttal Summary for Area Chair**

Dear Area Chair,

We appreciate the time and effort you and the reviewers have put into our submission. To help you quickly catch up, we've summarized the key feedback, along with our major revisions and additional experiments.

---

## **Overview of Assessments**

The paper received initial scores of **8 (KF7H), 6 (EZSh), 6 (FTxR), and 4 (bvZV)**. We are pleased that the reviewers recognized our method as a “novel, user-friendly, and highly efficient” pipeline (KF7H), with the “impressive” results that “outperform existing generation-based approaches” (FTxR, EZSh). The reviewers also highlighted the system’s “practical” nature and “controllability” (EZSh, bvZV).

---

## **Summary of Revisions and Additional Experiments**

To address the remaining concerns, we have conducted additional experiments and revised the manuscript accordingly. We believe all the issues have been resolved, as confirmed by Reviewer KF7H before the data leakage incident. All revisions have been highlighted in blue. We have also updated the supplementary videos, provided as individual MP4 files for each new addition. Below, we summarize the main, substantive revisions addressing the reviewers' core questions.

- **MLLM’s Performance on Challenging Materials** [EZSh/W2,Q1]: We have conducted additional qualitative experiments on diverse real-world images with uncommon/composed materials and unusual textures. The results are included in Appendix B.3 (Lines 1079-1120, MLLM Robustness on Challenging Materials) and Figure 13.

- **Comparison with Full Physics-based Simulation** [EZSh/W3,Q2; KF7H/Q1]: We have evaluated how our framework’s flame simulation compares to expert-level VFX software Blender. The full experimental setup and visual comparisons have been added in Appendix A.2 (Lines 909-959), Figure 11 and the supplementary video (comp_with_blender.mp4).

- **Improved Fire Rendering** [EZSh/W4; bvZV/W1,Q1]: We extended our lighting model to include temporal fluctuations in emitted light intensity. Details of the implementation have been added in the main paper (Lines 353-356), and the corresponding experimental results are included in the supplementary video (dynamic_lighting.mp4). Furthermore, to better demonstrate our rendering capabilities, we presented more pronounced specular effects (see Appendix A.3 Lines 984-1002, Figure 12 and supplementary video reflection.mp4).

- **Voxel Consumption** [FTxR/Q2]: We have additionally experimented with allowing the fire to consume material in the voxel, presented in Figure 16 in the Appendix. We further discussed this point in Appendix C (Lines 1242-1257, Material Degradation and Mass Loss).

- **Geometry Constraint for Fire Simulation** [bvZV/W2,Q1]: We clarified our ability to handle geometry constraints by adding descriptions at Lines 215-237 and conducted an additional experiment at Appendix A.2 Lines 900-902, Figure 9 and supplementary video (add_brick.mp4).

- **MLLM Comparison** [KF7H/Q2]: We have updated Appendix B.3 (Lines 1069-1077, Choice of MLLM) and Table 5 in the revised paper to include additional comparative results with Qwen3-VL-Plus.

---

Thank you again for the time and care you will devote to evaluating our submission. We sincerely hope this summary assists you in assessment of our work and the substantive improvements made during the rebuttal phase.


Best regards,

Authors

---

### Meta-Review · Area_Chair_ctXJ · 2026-01-05

**Summary:**

FieryGS introduces a practical framework for synthesizing physically grounded fire in 3D scenes by integrating MLLM-based material reasoning with 3DGS. The reviewers praised the system's efficiency, simplicity, and user-controllability. While initial concerns focused on the reliability of MLLMs for material estimation and the realism of simplified physics, the authors’ rebuttal includes comparisons with expert VFX tools (Blender) and robustness tests on complex materials, which could effectively resolve these issues. The work is a technically sound and high-utility contribution to 3D scene editing. Thus, I recommend the acceptance of the paper. The authors should ensure that the additional qualitative results on uncommon materials and the comparison with expert-level VFX tools (Blender) are fully integrated into the final manuscript.

**Reviewer Concerns:**

The authors have replied to all comments from the reviewers. The major concerns were addressed through the following updates:
- MLLM material reasoning: the authors provided new qualitative results in Appendix B.3 demonstrating that GPT-4o successfully handles uncommon/composite materials and unusual textures. Also, results from Qwen3-VL-Plus were provided.
- Visual realism and comparison with physically-based methods: The authors added a direct comparison with Blender's fire simulator, showing that their method achieves a significant speedup while maintaining comparable visual quality.
- Geometry constraints: concerns regarding fire interacting with scene geometry were addressed by adding an additional experiment in Appendix A.2, Figure 9, and supplementary video.

**Reviewer Scores:**

The manuscript received initial review scores of (8, 6, 6, 4).  During the discussion phase, Reviewer KF7H maintained a high score of 8, expressing satisfaction with the MLLM comparisons. Other reviewers did not engage in the discussion phase.

Reviewers EZSh and FTxR are experts in CG and acknowledged the impressive results and the utility of the practical pipeline, giving it an initial score of 6. I would assume they would tend to maintain the initial score given the MLLM material and visual realism concerns/ Reviewer bvZV gave a relatively short review with an initial score of 4 and I am not sure whether would turn to the positive side. Thus, I would assume an average score of 6-6.5. Based on the comments from the reviewers and the authors' replies, I recommend accepting the paper.

---

### Decision · Program_Chairs · 2026-01-26

Accept (Poster)